# DORAEMON: Decentralized Ontology-aware Reliable Agent with Enhanced Memory Oriented Navigation

## Abstract

Adaptive navigation in unfamiliar environments is crucial for household service robots but remains challenging due to the need for both low-level path planning and high-level scene understanding. While recent vision-language model (VLM) based zero-shot approaches reduce dependence on prior maps and scene-specific training data, they face significant limitations: spatiotemporal discontinuity from discrete observations, unstructured memory representations, and insufficient task understanding leading to navigation failures. We propose DORAEMON (Decentralized Ontology-aware Reliable Agent with Enhanced Memory Oriented Navigation), a novel cognitive-inspired, zero-shot, end-to-end framework consisting of Ventral and Dorsal Streams that mimics human navigation capabilities. The Dorsal Stream implements the Hierarchical Semantic-Spatial Fusion and Topology Map to handle spatiotemporal discontinuities, while the Ventral Stream combines CoDe-VLM and Exec-VLM to improve decision-making. Our approach also develops Nav-Ensurance to ensure navigation safety and efficiency. We evaluate DORAEMON on the HM3Dv1, HM3Dv2, MP3D, where it achieves state-of-the-art performance on both SR and SPL metrics, significantly outperforming existing methods. We also introduce a new evaluation metric (AORI) to assess navigation intelligence better. Comprehensive experiments demonstrate DORAEMON's effectiveness in zero-shot and end-to-end navigation without requiring prior map building or pre-training. Our code is available at https://anonymous.4open.science/r/DORAEMON-8D4D.

## 1 Introduction

Adaptive navigation in complex and unseen environments (Batra et al., 2020) is a key capability for household service robots. This task requires robots to move from a random starting point to the location of a target object without any prior knowledge of the environment. For humans, navigation appears almost trivial; however, navigation remains a highly challenging problem for robots: it demands not only low-level path planning to avoid obstacles and reach the destination, but also high-level scene understanding to interpret and make sense of the surrounding environment.

Most existing navigation methods rely on the construction of prior maps(Cadena et al., 2017)

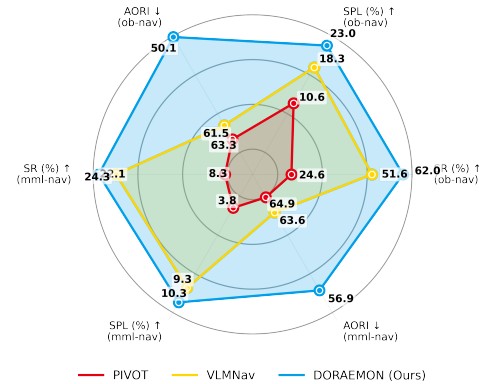

Figure 1: Performance comparison of end-to-end methods in object navigation(ob-nav) and multi-modal lifelong navigation(mml).

or require extensive scene-specific data for task-oriented pre-training(Szot et al., 2021). Recently, some works(Yin et al., 2024; Zhong et al., 2024; Wu et al., 2024) have begun to explore zero-training and zero-shot navigation strategies. By relying on textual descriptions of the current task, image inputs, and previously observed historical information, these approaches achieve navigation without dependence on environment or task-specific data, gradually shedding the reliance on scene priors.

Although zero-shot and zero-training navigation methods offer a novel perspective, they still face numerous challenges in practical applications. On the one hand, most current navigation methods are non-end-to-end, where the agent's spatial actions are mapped to a discrete set. These discrete actions result in paths that are neither smooth nor efficient. To align with a target, the agent may require multiple small-angle rotations. On the other hand, the primary bottleneck for current Vision-Language Models (VLMs) in long-range navigation is their inadequate memory mechanisms. Their reliance on discrete observational inputs prevents a cohesive understanding of spatiotemporal continuity. More critically, the prevalent approaches(Ramakrishnan et al., 2024; Nasiriany et al., 2024) of storing history as an unstructured log within a single-step decision paradigm fundamentally compromises their ability to perform effective long-term path planning.

Even though end-to-end methods like VLMnav(Nasiriany et al., 2024; Goetting et al., 2024) utilize historical information, they typically store this information in a flat, unstructured manner, which fundamentally limits their ability to perform long-range navigation. Additionally, VLMs' frequent insufficient understanding of task semantics often leads to poor decision-making, and the lack of reliable check mechanisms for navigation states frequently results in unreliable behaviors such as spinning in place during navigation tasks.

Inspired by cognitive science "Decentralized Ontology" principles (Bouquet et al., 2004), we propose the Decentralized Ontology-aware Reliable Agent with Enhanced Memory Oriented Navigation (DORAEMON), which consists mainly of a Ventral Stream and a Dorsal Stream. The core theoretical premise is that knowledge is inherently distributed and context-dependent, composed of multiple local perspectives, rather than a single, monolithic world model. Specifically, DORAEMON instantiates two specialized knowledge centers: the Ventral Stream establishes a static Task Ontology for instruction compilation, while the Dorsal Stream maintains a dynamic Spatial Ontology for environmental mapping. The Dorsal Stream addresses spatio-temporal discontinuities through a Topology Map and a Hierarchical Semantic-Spatial Fusion, which employs a Periodic Global Reconstruction strategy to refresh the memory hierarchy from bottom-up observations, effectively mitigating stale information and graph explosion. Additionally, the Ventral Stream improves task understanding by utilizing a CoDe-VLM (Compositional Decomposition VLM) and Exec-VLM (Execution VLM) for navigation. Furthermore, DORAEMON features a Nav-Ensurance system that enables the agent to autonomously detect and respond to abnormal conditions, such as becoming stuck or blocked during navigation. To evaluate navigation performance more comprehensively, we propose a new metric called the Adaptive Online Route Index (AORI). Fig 2 conceptually illustrates limitations of traditional VLN methods and contrasts them with DORAEMON.

In summary, the main contributions of this work are:

- We propose DORAEMON, a novel adaptive navigation framework inspired by cognitive principles of decentralized knowledge, consisting of ventral and Dorsal Streams, enabling end-to-end and zero-shot navigation in completely unfamiliar environments without pre-training, offering plug-and-play compatibility with any VLMs.

- We propose the Dorsal Stream, which involves designing a Topology Map and a Hierarchical Semantic-Spatial Fusion Network to effectively manage spatio-temporal discontinuities. Crucially, we introduce a periodic global reconstruction mechanism to ensure memory freshness and robustness. Additionally, we introduce the Ventral Stream, incorporating a synergistic reasoning component that combines CoDe-VLM for understanding ontological tasks and Exec-VLM for enhanced task comprehension and planning.

- We develop Nav-Ensurance, which includes multi-dimensional stuck detection and context-aware escape mechanisms. We propose a new evaluation metric called AORI to quantify the efficiency of the agents exploration. Our method demonstrates state-of-the-art performance across various navigation tasks.

## 2 RELATED WORK

### 2.1 ZERO-SHOT NAVIGATION

Navigation methods are broadly supervised or zero-shot. Supervised approaches train visual encoders with reinforcement/imitation learning (Khandelwal et al., 2022; Maksymets et al., 2021;

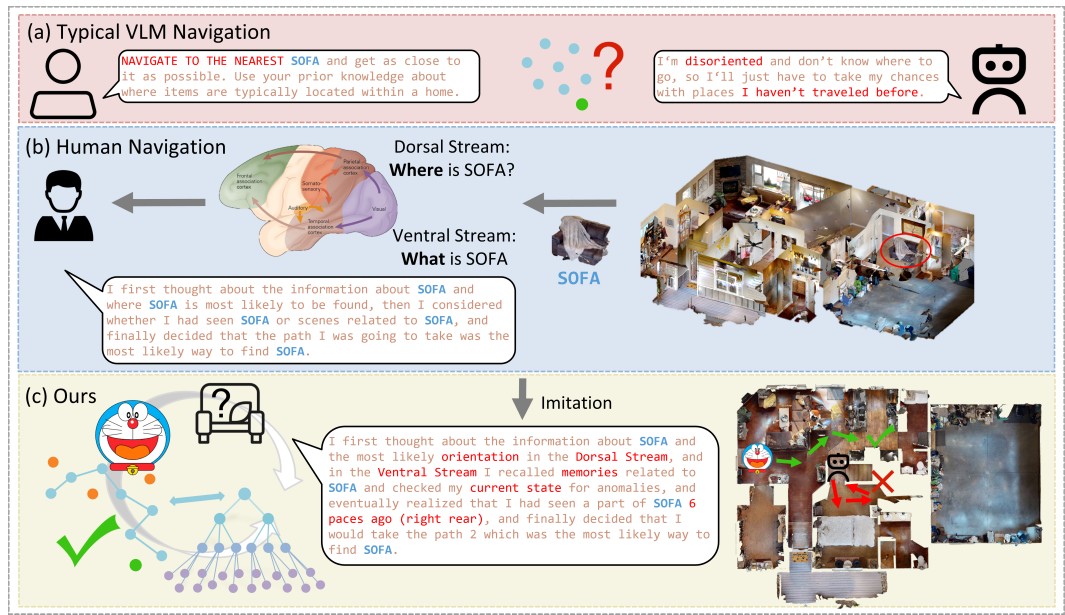

Figure 2: (a) Illustrates limitation of typical VLM navigation (red arrow). (b) DORAEMON's cognitive inspiration from human navigation. (c) Our DORAEMON method.

Ramrakhya et al., 2022; Chen et al., 2022) or build semantic maps from training data (Zhang et al., 2025a; Min et al., 2021; Zheng et al., 2022), struggling with novel scenarios due to data dependency. Zero-shot methods address this using open-vocabulary understanding, increasingly leveraging foundation models like LLMs and VLMs. LLMs provide commonsense reasoning via object-room correlation (Yin et al., 2024; Zhou et al., 2023; Wu et al., 2024), semantic mapping (Yu et al., 2023), and chain-of-thought planning (Cai et al., 2025; Yin et al., 2024; Shah et al., 2023a), though methods like NavGPT (Zhou et al., 2024) relying solely on text often lack geometric precision. While VLMs align visual observations with textual goals, these foundation model-guided techniques include image-based methods mapping targets to visual embeddings (Wen et al., 2025; Gadre et al., 2023; Al-Halah et al., 2022)distinct from NavigateDiff (Qin et al., 2025) which uses indirect generative prediction and map-based approaches using frontier (Zhong et al., 2024; Zhang et al., 2025a; Chen et al., 2023; Kuang et al., 2024; Yu et al., 2023; Shah et al., 2023a) or waypoint-based maps (Wu et al., 2024) with LLM/VLM reasoning. VLM-based strategies either use VLMs for recognition with traditional planning and extra perception models (Rahmanzadehgervi et al., 2024; Zhang et al., 2025b), or, like PIVOT (Nasiriany et al., 2024) and VLMnav (Goetting et al., 2024), directly produce actions end-to-end via visual prompting. Unlike NaVid (Zhang et al., 2024a) which requires continuous video, DORAEMON handles spatiotemporal discontinuity even with discrete observations. Despite progress, many zero-shot methods, especially those processing observations independently, face challenges integrating temporal information and handling complex spatial reasoning in unfamiliar environments.

## 2.2 MEMORY MECHANISMS IN NAVIGATION

Memory representations in navigation systems have evolved through various architectures, including episodic buffers that maintain observation sequences (Goetting et al., 2024; Shah et al., 2023b; Hsu et al., 2022), spatial representations prioritizing geometric information (Zhong et al., 2024; Zhang et al., 2025b), graph-based semantic structures capturing object relationships (Yin et al., 2025; 2024)represented by recent works like Mem4Nav (He et al., 2025) and TopoNav (Hossain et al., 2024) , predictive world models attempting to forecast environmental states (Cao et al., 2024; Nie et al., 2025) and the memory capacity acquired through training(Zhu et al., 2025). These systems typically process semantic and spatial information separately, with limited integration between perception and reasoning modules. Most approaches focus on either building representations or enhancing reasoning mechanisms independently. Differently, DORAEMON integrates these aspects

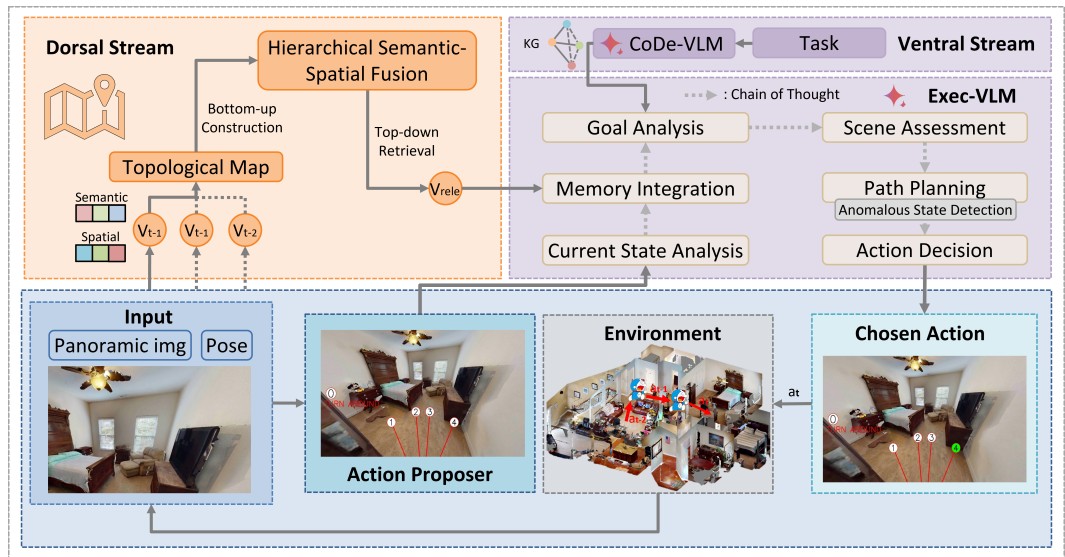

Figure 3: Workflow of DORAEMON: From instruction input to continuous action output. The Ventral Stream compiles the task into a static Task Ontology, while the Dorsal Stream builds a dynamic Spatial Ontology. Exec-VLM integrates these distinct ontologies for decision-making.

through a hierarchical semantic-spatial fusion network with bidirectional information flow between ventral and dorsal processing streams, utilizing a periodic global reconstruction strategy to prevent graph explosion and stale nodes common in incremental updates.

## 2.3 COGNITIVE NEUROSCIENCE INSPIRATION IN NAVIGATION

Navigation systems are influenced by cognitive neuroscience; recent models like CogNav(Cao et al., 2024) and BrainNav(Ling & Qianqian, 2025) incorporate cognitive elements, but they do not fully embody Decentralized Ontology. CogNav utilizes a finite state machine for cognitive states, but may have limitations in knowledge integration. BrainNav mimics biological functions but doesn't deeply engage in decentralized information processing. In contrast, DORAEMON is inspired by Decentralized Ontology(Bouquet et al., 2004), which suggests that human knowledge is organized through interconnected cognitive systems that enable context-dependent reasoning. It emphasizes the integration and bidirectional exchange of information between Dorsal Stream and Ventral Stream, allowing for the construction of semantic relationships that enhance spatial understanding and support flexible, context-aware navigation.

## 3 METHODS

**Task Formulation** We address the Navigation task (Batra et al., 2020), where an agent, starting from an initial pose, must locate and navigate to a target within a previously unseen indoor environment. At step $t$, the agent receives observation $I_t$, current pose $P_t$ and a task specification $T$, which can be either a simple object category (e.g., "sofa") or an instruction (e.g., "find the red chair" or"the plant on the desk") for tasks like GOAT (Khanna et al., 2024). Based on these inputs, the agent must decide on an action $a_t$. While many prior works utilize a discrete action space, our end-to-end framework employs a continuous action representation in polar coordinates $(r_t, \theta_t)$, where $r_t$ specifies the forward distance to move, and $\theta_t$ denotes the change in orientation. Crucially, the action space also includes a stop action. The task is considered successful if the agent executes the stop action after meeting successive stop triggers in steps $t$ and $t + 1$. The trigger occurs when 1) the agent is within a predefined distance threshold $d_{success}$ of the target object; 2) the target object is visually confirmed within the agent's current observation $I_t$.

**Methods Overview** Our DORAEMON framework achieves end-to-end and zero-shot navigation through two decentralized cognitive-inspired streams, as depicted in Figure 3. Consistent with the

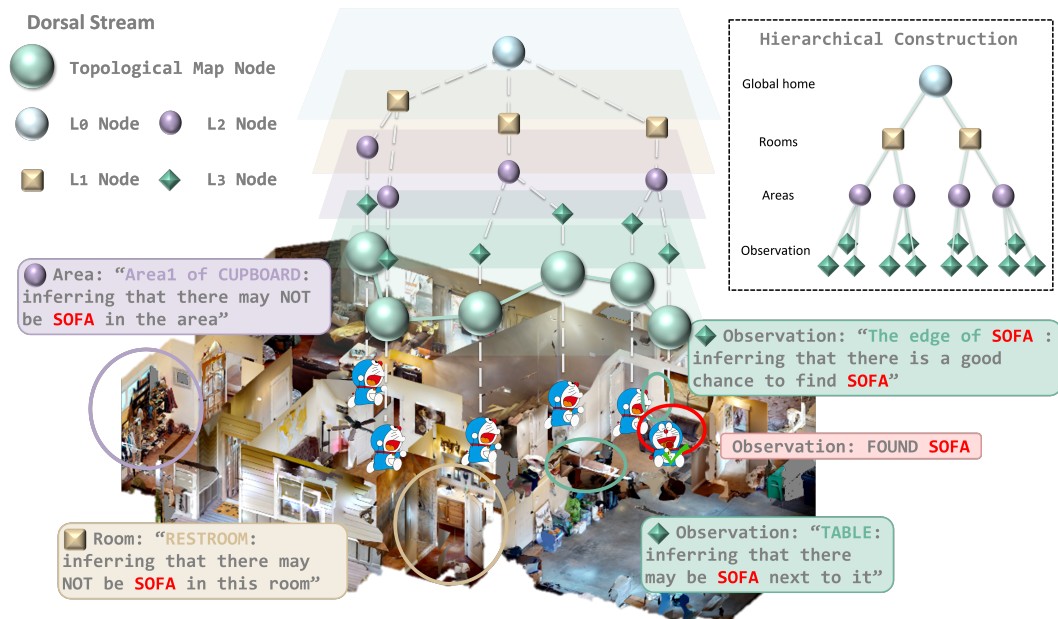

Figure 4: Architecture of Topological Map and Hierarchical Construction built in Dorsal Stream for spatio-temporal memory. The top view in the middle shows the content of different nodes during navigation, and the upper right part represents the Hierarchical Construction of a node.

Decentralized Ontology principle, we instantiate two specialized knowledge centers: the Ventral Stream constructs a Task Ontology for semantic understanding, while the Dorsal Stream maintains a Spatial Ontology for environmental mapping. Given an input with a panoramic image $I_t$ and a pose $P_t$ at step $t$, they are processed by Action Proposer (Appendix B) and Dorsal Stream(Section 3.1), respectively. In the Action Proposer, a candidate image $I_{anno}^t$ is generated with a set of action candidates $A_{final}^t$. Concurrently, the Dorsal Stream extracts semantic and spatial information from $I_t$ using Hierarchical Semantic-Spatial Fusion and stores it within the Topology Map as node $v_t$. The relevant node $v_{rele}$ can be accessed by up-down retrieval. After that, $v_{rele}$ and $I_{anno}^t$ are input to the Exec-VLM to select the best action based on the given information(Section 3.2.2). At the same time, the Exec-VLM receives a task-specific knowledge graph (KG) relevant to the task $T$, which is generated by the CoDe-VLM (Section3.2.1) in the Ventral Stream (Section 3.2). The Exec-VLM integrates the information through chain of thought (Appendix I), identifies abnormal conditions (Section 3.3), and outputs the final action $a_t$. The agent performs this action $a_t$ in the environment, navigates, and makes the next decision at step $t + 1$.

## 3.1 DORSAL STREAM

The Dorsal Stream, similar to the "where/how" pathway in cognition, is responsible for processing the spatial information to effectively navigate. As illustrated in Figure 4, at each step $t$, the agent constructs $v_k$ on the Topology Map (Section 3.1.1). Subsequently, the Hierarchical Semantic-Spatial Fusion (Section 3.1.2) organizes the information into a hierarchical structure from the bottom up. Unlike incremental-only approaches, we employ a Periodic Global Reconstruction mechanism here to ensure the map remains robust against stale nodes and structural drifts.

### 3.1.1 TOPOLOGICAL MAP

The topological map $\mathcal{G} = (\mathcal{V}, \mathcal{E})$ is constructed incrementally. Each node $v_t \in \mathcal{V}$ encapsulates the agent's state at timestep $t$, defined as a tuple $(p_t, q_t, I_t, L_t, o_t, s_t)$. $p_t$ and $q_t$ denote the agent's position and orientation, which constitute the pose $P_t$. $I_t$ is the visual observation, $L_t$ is its corresponding language description, $o_t$ is the target likelihood estimation, and $s_t$ represents a semantic embedding of the observation (e.g., using lightweight CLIP-ViT-B/32 features). A new node $v_{new}$ is added to $\mathcal{V}$ based on spatio and temporal criteria: a new node is created if either the time elapsed

since the last node addition $t_{\text{curr}} - t_{\text{prev}}$ exceeds a temporal threshold $S_{\text{update}}$, or if the agent's Euclidean distance from the previous node $\|p_{\text{curr}} - p_{\text{prev}}\|_2$ surpasses a spatial threshold $\delta_{\text{sample}}$. Upon its creation, $v_{\text{new}}$ is connected to its predecessor node $v_{\text{prev}}$ via a new edge. Note that navigability checks during this process rely solely on depth map analysis without heavy external segmentation models.

### 3.1.2 HIERARCHICAL SEMANTIC-SPATIAL FUSION

**Hierarchical Construction.** Building upon the multimodal information encapsulated in the Topological Map nodes $v_t \in \mathcal{V}$, our module organizes this data into a structured SemanticForest. The nodes $h_j$ within this hierarchy are formally defined as:

$$h_j = \Big( id_j, \ l_j, \ \mathcal{P}_j, \ \mathcal{C}_j \Big), \tag{1}$$

where $id_j$ is a unique identifier, $l_j \in \{L_0, L_1, L_2, L_3\}$ denotes the semantic level, and $\mathcal{P}_j, \mathcal{C}_j$ represent the sets of parent and child node references, respectively. The hierarchy categorizes nodes into four distinct semantic levels: $L_3$ (Observation, directly anchored to topological nodes $v_t$), $L_2$ (Area, representing clusters of observations), $L_1$ (Room, representing functional spaces), and $L_0$ (Environment, the global root).

Unlike traditional topological approaches that rely solely on incremental updateswhich often lead to redundant nodes and error accumulationwe implement a **Periodic Global Reconstruction** mechanism. Specifically, at every update interval $S_{update}$, the system temporarily freezes the current set of bottom-level leaf nodes ($L_3$) and discards the existing high-level structures ($L_2, L_1, L_0$). It then performs a fresh, global clustering operation on all $L_3$ nodes to regenerate areas and rooms based on the latest accumulated evidence. This "clean-slate" strategy allows the agent to automatically rectify early semantic misclassifications (e.g., re-merging a fragmented "kitchen" area) as more comprehensive visual context becomes available, effectively resolving the issue of stale nodes.

While the reconstruction creates the hierarchy bottom-up ($L_3 \rightarrow L_2 \rightarrow L_1 \rightarrow L_0$), the clustering logic is adaptive. For instance, the clustering distance threshold $\delta_{cluster}$ used to form $L_2$ nodes is not a static hyperparameter; instead, it is dynamically adjusted based on the density and total count of active leaf nodes. This adaptive mechanism maintains a balanced tree structure, preventing the "graph explosion" problem often encountered in large-scale environment exploration (see Appendix G for detailed algorithm).

**Hierarchical Memory Retrieval.** To efficiently retrieve relevant information (e.g., locating "sofa" within the memory), the system employs a top-down heuristic search. This search is guided by a multi-factor scoring function $S(h_i)$, evaluated at each node $h_i$ during traversal:

$$S(h_i) = \alpha_{\text{sem}} S_{\text{sem}}(h_i, T) + \alpha_{\text{spa}} S_{\text{spa}}(h_i) + \alpha_{\text{key}} S_{\text{key}}(h_i, T) + \alpha_{\text{time}} S_{\text{time}}(h_i), \tag{2}$$

where $S_{\text{sem}}$ computes the semantic embedding similarity between node $h_i$ and the task query $T$, and $S_{\text{spa}}$ measures proximity to the current position using an exponential decay function. $S_{\text{key}}$ evaluates keyword overlap, and $S_{\text{time}}$ prioritizes recent observations. The weights $\alpha$ balance these components, ensuring the retrieved memory is both semantically relevant and spatially/temporally accessible (see Appendix H for calculation details).

### 3.2 VENTRAL STREAM

The Ventral Stream, analogous to the "what" pathway in human cognition, integrates two key components: CoDe-VLM (Compositional Decomposition VLM, Section 3.2.1) and Exec-VLM (Execution VLM, Section 3.2.2). Unlike prior models that encode task information into a single, entangled vector, our architecture explicitly disentangles task comprehension from execution. This decentralized design mirrors the Ventral Stream's approach to compositional understanding, first compiling knowledge and then acting upon it.

### 3.2.1 CoDe-VLM: Compiling Tasks into Knowledge Graphs

To build a deep and structured understanding of the task, CoDe-VLM acts as a semantic compiler. It leverages the vast world knowledge embedded within a VLM to on-the-fly compile an unstructured instruction $T$ into a dynamic, task-specific knowledge graph (KG).

CoDe-VLM generates a graph structure encapsulating nodes and relational edges. This task KG, formed from extracted semantic attributes like general description, appearance, and location, constitutes our explicit and compositional representation of the task. This representation not only enables the agent to robustly verify objects encountered during navigation but also provides crucial priors for planning by interfacing with the spatial reasoning components of the Dorsal Stream.

### 3.2.2 Exec-VLM: Executing Actions via Graph-based Reasoning

The Exec-VLM serves as the agent's executive core, responsible for determining the optimal action by combining visual observations, spatial awareness from Dorsal Stream, and the structured task semantics provided by CoDe-VLM. Crucially, instead of making decisions in a high-dimensional, entangled feature space, Exec-VLM performs explicit reasoning on the task knowledge graph.

We steer this reasoning process using Chain-of-Thought (CoT). The CoT guides Exec-VLM to break down the complex navigation task into interpretable sub-steps: current state analysis, memory integration, goal analysis, scene assessment, path planning, and action decision. During the "goal analysis" step, for instance, the model directly queries the nodes and edges of the KG to confirm the target's identity and properties, rather than relying on a fragile memory of the initial instruction.

## 3.3 Nav-Ensurance

To enhance the evaluation of safety and efficiency in navigation, we present a new metric Area Overlap Redundancy Index (AORI) (Section 3.3.1). Additionally, we develop Nav-Ensurance, including Multimodal Stuck Detection (Section 3.3.2), context-aware escape strategies (Section 3.3.3), and adaptive precision navigation (Section 3.3.4) to ensure navigation systems reliably and effectively.

### 3.3.1 Area Overlap Redundancy Index (AORI)

We introduce the Area Overlap Redundancy Index (AORI) to quantify the efficiency of the agent's navigation strategy by measuring overlap in area coverage. A high AORI indicates excessive path overlap and inefficient exploration, specifically addressing the limitations of conventional coverage metrics that neglect temporal-spatial redundancy. AORI is formally defined as:

$$\text{AORI} = 1.0 - (w_c \cdot (1.0 - r_{\text{overlap}})^2 + w_d \cdot (1.0 - d_{\text{norm}})), \tag{3}$$

Where $r_{\text{overlap}}$ represents the ratio of revisited areas, $d_{\text{norm}}$ is the normalized density, and $w_c = 0.8, w_d = 0.2$ are weighting coefficients. Further details are provided in Appendix E.

### 3.3.2 Multimodal Stuck Detection

To detect if it is stuck, the agent analyzes its trajectory over a sliding window of $T$ steps by computing two key metrics: the progress efficiency $\eta$ and the rotational-to-translational ratio $\rho$.

$$\eta = \frac{\|p_T - p_0\|_2}{\sum_{t=1}^{T} \|p_t - p_{t-1}\|_2}, \quad \rho = \frac{\sum_{t=1}^{T} |\theta_t - \theta_{t-1}|}{\sum_{t=1}^{T} \|p_t - p_{t-1}\|_2}. \tag{4}$$

A stuck state is confirmed if a weighted score $S = w_\eta \cdot \mathbb{I}[\eta < \tau_\eta] + w_\rho \cdot \mathbb{I}[\rho > \tau_\rho]$ remains above a threshold $S_{\text{th}}$ for $k$ consecutive windows. These metrics effectively detect situations where the agent makes little forward progress (low $\eta$) or is spinning in place (high $\rho$).

### 3.3.3 Context-aware Escape Strategies

When a stuck state is detected, the system selects an appropriate escape strategy based on the perceived information from Dorsal Stream(Section 3.1). For instance, in corner traps (perceived dead

ends), a large turn is executed. In narrow passages, a small backward step followed by a randomized direction change is employed. If the environmental context is ambiguous, the agent will analyze recent successful movement directions and attempt to move perpendicularly, significantly improving escape capabilities from complex trap situations.

### 3.3.4 ADAPTIVE PRECISION NAVIGATION

As the agent nears the target object, it will activate a precision navigation mode. In this mode, the distance component $d$ of all proposed actions $(d, \theta)$ is scaled down by a factor $\gamma_{\text{step}}$ to enable fine-grained positioning adjustments:

$$a_{\text{precise}} = (d \cdot \gamma_{\text{step}}, \theta) \quad \text{for action } (d, \theta) \in A_{\text{actions}}. \tag{5}$$

Additionally, when activating the precision navigation mode, the system can utilize visual analysis (using VLM) to create more detailed action options, thereby maximizing final positioning accuracy.

## 4 EXPERIMENTS

**Datasets** We evaluate our proposed DORAEMON within the Habitat simulator (Savva et al., 2019) on four large-scale datasets: HM3Dv1(Ramakrishnan et al., 2021)(Object Navigation), HM3Dv2(Yadav et al., 2023)(Object Navigation), and MP3D (Chang et al., 2017)(Object Navigation), GOAT(Khanna et al., 2024) (Multi-modal lifelong navigation, using HM3Dv2).

**Implementation Details and Evaluation Metrics** The action space includes `stop`, `move_forward` where the distance parameter is sampled from the continuous range $[0.5\text{m}, 1.7\text{m}]$, and `rotate`. We adopt standard metrics to evaluate navigation performance: Success Rate (SR), the percentage of episodes where the agent successfully stops near a target object; Success weighted by Path Length (SPL), defined as $\frac{1}{N} \sum_{i=1}^{N} S_i \frac{l_i}{\max(p_i, l_i)}$, rewarding both success and efficiency; and our proposed Area Overlap Redundancy Index (AORI) (Equation equation 3), which quantifies navigation by penalizing redundant exploration (lower is better). More information is set in the Appendix F.

**Baselines** We compare DORAEMON against several state-of-the-art navigation methods on the HM3Dv2, HM3Dv1, and MP3D. Our main comparison focuses on end-to-end approaches. Beyond these direct end-to-end counterparts, we also consider a broader set of recent methods for non-end-to-end object navigation methods. More baseline details are set in the Appendix J.

### 4.1 METHODS COMPARISON

**End-to-end Methods:** We evaluate our approach on the HM3Dv2 (ObjectNav,val, Table 1 (a)) and HM3Dv1(GOAT, val, Table 1 (b)) with other end-to-end baselines. DORAEMON achieves state-of-the-art performance on both datasets, outperforming other methods by a significant margin.

Table 1: Comparison of end-to-end navigation methods on different benchmarks.

(a) ObjectNav benchmark

| Method | SR (%) ↑ | SPL (%) ↑ | AORI (%) ↓ |
|---|---|---|---|
| Prompt-only | 29.8 | 0.107 | - |
| PIVOT(Nasiriany et al., 2024) | 24.6 | 10.6 | 63.3 |
| VLMNav(Goetting et al., 2024) | 51.6 | 18.3 | 61.5 |
| **DORAEMON (Ours)** | **62.0** | **23.0** | **50.1** |
| **Improvement** | **20.2** | **10.0** | **18.5** |

(b) GOAT benchmark

| Method | SR (%) ↑ | SPL (%) ↑ | AORI (%) ↓ |
|---|---|---|---|
| Prompt-only | 11.3 | 3.7 | - |
| PIVOT(Nasiriany et al., 2024) | 8.3 | 3.8 | 64.9 |
| VLMNav(Goetting et al., 2024) | 22.1 | 9.3 | 63.6 |
| **DORAEMON (Ours)** | **24.3** | **10.3** | **56.9** |
| **Improvement** | **10.0** | **10.8** | **10.5** |

**Non-end-to-end methods:** Most methods are non-end-to-end, their reliance on fine-grained discrete actions is a significant departure from natural human behavior, underscoring the superiority of an end-to-end approach. To ensure a fair comparison with these methods that utilize a discrete action set $\mathcal{A}$: `move forward` $0.25\text{m}$, `turn left/turn right` $30°$, `look up/lookdown` $30°$, `stop`, and a common 500 steps episode limit, we conduct an additional set of experiments. In these, we normalize our agent's interactions by approximating an equivalent number of standard

discrete steps for each of DORAEMON's actions. During our experiments, one DORAEMON step $t$ was equivalent to about 9-10 non-end-to-end step $t_n$. Note that since the SPL metric explicitly accounts for path length, our superior SPL performance confirms that the gains are due to efficient planning, not merely larger step sizes.

Table 2: Comprehensive comparison with state-of-the-art methods on ObjectNav benchmarks. TF refers to training-free, ZS refers to zero-shot, and E2E refers to end-to-end.

| Method | ZS | TF | E2E | HM3Dv1 | | HM3Dv2 | | MP3D | |
|--------|----|----|-----|--------|--------|--------|--------|--------|--------|
| | | | | SR(%) ↑ | SPL(%) ↑ | SR(%) ↑ | SPL(%) ↑ | SR(%) ↑ | SPL(%) ↑ |
| ProcTHOR (Deitke et al., 2022) | × | × | × | 54.4 | **31.8** | - | - | - | - |
| SemEXP (Chaplot et al., 2020) | ✓ | × | × | - | - | - | - | 36.0 | 14.4 |
| Habitat-Web(Ramrakhya et al., 2022) | ✓ | × | × | 41.5 | 16.0 | - | - | 31.6 | 8.5 |
| PONI (Ramakrishnan et al., 2022) | ✓ | × | × | - | - | - | - | 31.8 | 12.1 |
| ProcTHOR-ZS (Deitke et al., 2022) | ✓ | × | × | 13.2 | 7.7 | - | - | - | - |
| ZSON (Majumdar et al., 2022) | ✓ | × | × | 25.5 | 12.6 | - | - | 15.3 | 4.8 |
| PSL (Sun et al., 2024) | ✓ | × | × | 42.4 | 19.2 | - | - | - | - |
| Pixel-Nav (Cai et al., 2024) | ✓ | × | × | 37.9 | 20.5 | - | - | - | - |
| SGM (Zhang et al., 2024b) | ✓ | × | × | **60.2** | 30.8 | - | - | 37.7 | 14.7 |
| ImagineNav (Zhao et al., 2024) | ✓ | × | × | 53.0 | 23.8 | - | - | - | - |
| CoW (Gadre et al., 2023) | ✓ | ✓ | × | - | - | - | - | 7.4 | 3.7 |
| ESC (Zhou et al., 2023) | ✓ | ✓ | × | 39.2 | 22.3 | - | - | 28.7 | 14.2 |
| L3MVN (Yu et al., 2023) | ✓ | ✓ | × | 50.4 | 23.1 | 36.3 | 15.7 | 34.9 | 14.5 |
| VLFM (Yokoyama et al., 2024) | ✓ | ✓ | × | 52.5 | 30.4 | 63.6 | **32.5** | 36.4 | **17.5** |
| VoroNav (Wu et al., 2024) | ✓ | ✓ | × | 42.0 | 26.0 | - | - | - | - |
| TopV-Nav (Zhong et al., 2024) | ✓ | ✓ | × | 52.0 | 28.6 | - | - | 35.2 | 16.4 |
| InstructNav (Long et al., 2024) | ✓ | ✓ | × | - | - | 58.0 | 20.9 | - | - |
| SG-Nav (Yin et al., 2024) | ✓ | ✓ | × | 54.0 | 24.9 | 49.6 | 25.5 | 40.2 | 16.0 |
| **DORAEMON (Ours)** | ✓ | ✓ | ✓ | 55.6 | 21.4 | **66.5** | 20.6 | **41.1** | 15.8 |

**Non-end-to-end methods:** Most methods are non-end-to-end; their reliance on fine-grained discrete actions is a significant departure from natural human behavior. To ensure a fair comparison with methods utilizing the standard discrete action set $\mathcal{A}$ and a common 500 steps episode limit, we normalize our agent's interactions. As validated in Table 3, one DORAEMON step typically equates to 9-10 discrete steps, proving that our efficiency gains stem from superior planning.

Table 3: Step conversion analysis validating fairness. Left: Decomposition of single continuous actions. Right: Total steps in full episodes, showing a ∼1:10 efficiency ratio.

| (a) Single Action Breakdown | | (b) Full Episode Steps | |
|-----------------------------|--------------------------|------------------------|------------------|
| **Continuous Action** $(r, \theta)$ | **Discrete Equiv.** $(t_n)$ | **Ours** $(t)$ | **Discrete** $(t_n)$ |
| (1.27m, 53°) | 6×Fwd + 2×Turn (**8**) | 3 | 23 |
| (1.70m, 60°) | 7×Fwd + 2×Turn (**9**) | 16 | 95 |
| (1.10m, 93°) | 5×Fwd + 4×Turn (**9**) | 42 | 300 |

Compared to the non-end-to-end approach in the Table 2, DORAEMON achieves state-of-the-art performance on SR, despite normalizing our action to set $\mathcal{A}$. Each action performed by ours corresponds to several actions in this set. In fact, we only run about 60 end-to-end steps, which further demonstrates the excellence of our DORAEMON.(details are provided in the Appendix C)

**Ablation Studies.** We perform comprehensive ablation studies to validate our design choices, with results summarized in Table 4. (a) Core Components: Table 4(a) shows that each component is crucial. Removing both the Dorsal and Ventral streams severely degrades performance, confirming their synergistic effect. Disabling the Nav-Ensurance mechanism also notably worsens the AORI, highlighting its effectiveness in error prevention. (b) Choice of VLM: The VLM ablation in Table 4(b) indicates that while Gemini-1.5-Pro is optimal, our framework remains highly effective with smaller models (e.g., Qwen-7B achieves 49.5% SR, still surpassing baselines like PIVOT). This demonstrates our architecture's inherent strength independent of the VLM's capacity, suggesting future compatibility with evolving VLMs. (c) Hyperparameter Sensitivity: The analysis in Table 4(c) reveals a trade-off between metrics. For example, setting TopK=12 yields the highest SR but at the cost of SPL. Our default hyperparameters are carefully chosen to achieve a robust and balanced performance across all metrics, rather than over-optimizing for a single one.

Table 4: A comprehensive ablation study of DORAEMON across different datasets, including variations in modules, VLMs, and hyperparameters. All experiments were evaluated over 100 episodes. Our DORAEMON uses the default hyperparameters: TopK=8, memory update interval=3, area grid size=2.

| Method / Configuration | HM3Dv2 | | | HM3Dv1 | | | MP3D | | |
|---|---|---|---|---|---|---|---|---|---|
| | SR (%) ↑ | SPL (%) ↑ | AORI (%) ↓ | SR (%) ↑ | SPL (%) ↑ | AORI (%) ↓ | SR (%) ↑ | SPL (%) ↑ | AORI (%) ↓ |
| *(a) Ablation of different modules* | | | | | | | | | |
| w/o Dorsal & Ventral Stream | 51.6 | 18.3 | 61.5 | 48.4 | 18.9 | 53.7 | 38.8 | 13.9 | 64.3 |
| w/o Dorsal & CoDe-VLM | 54.0 | 19.8 | 59.1 | 51.2 | 19.4 | 52.5 | 40.2 | 14.2 | 63.8 |
| w/o Dorsal Stream | 59.0 | 22.7 | 56.3 | 53.8 | 20.5 | 51.1 | 40.9 | 14.6 | 65.1 |
| w/o Nav-Ensurance | 60.0 | 22.5 | 54.9 | 53.1 | 20.7 | **50.9** | 42.2 | 15.3 | 60.4 |
| *(b) Ablation of different VLMs (on HM3Dv2)* | | | | | | | | | |
| Qwen-7B | 49.5 | 20.6 | 68.7 | - | - | - | - | - | - |
| Gemini-1.5-Flash | 58.0 | 20.1 | 54.8 | - | - | - | - | - | - |
| Gemini-2-Flash | 59.0 | 21.5 | 57.9 | - | - | - | - | - | - |
| *(c) Ablation of Hyperparameters (on HM3Dv2)* | | | | | | | | | |
| w/ TopK = 12 | **65.0** | 22.57 | 42.89 | - | - | - | - | - | - |
| w/ TopK = 4 | 59.0 | 22.78 | 42.03 | - | - | - | - | - | - |
| w/ memory update interval = 1 | 53.0 | 19.94 | 45.54 | - | - | - | - | - | - |
| w/ memory update interval = 5 | 62.0 | 23.61 | 44.22 | - | - | - | - | - | - |
| w/ area grid size = 1 | 61.0 | 21.97 | **39.66** | - | - | - | - | - | - |
| w/ area grid size = 3 | 61.0 | 22.95 | 43.57 | - | - | - | - | - | - |
| **DORAEMON (Ours, default)** | 61.0 | **23.7** | 48.8 | **55.6** | **21.4** | **49.1** | 41.1 | **15.8** | 59.3 |

## 4.2 NAVIGATION IN REAL WORLD

To validate the Sim-to-Real generalization of our model, we deployed our DORAEMON in a novel office environment. Despite the significant domain gap, the agent successfully completed navigation tasks and robustly recovered from stuck states (e.g., getting blocked between sofas) using Nav-Ensurance strategies. Figure 5 shows a representative trial. More demos are available on our project homepage.

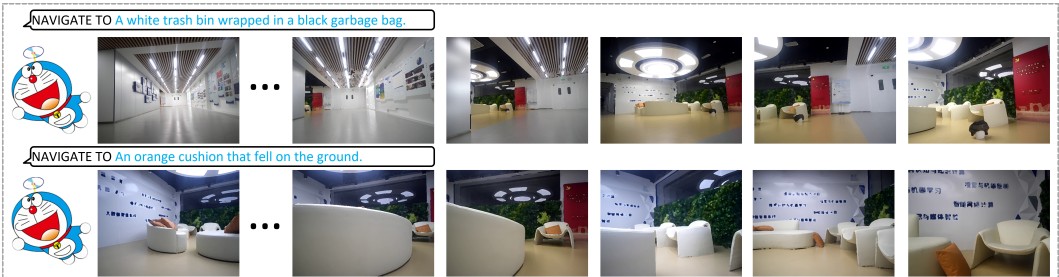

Figure 5: DORAEMON's Performance in sim2real

## 5 CONCLUSION

In this paper, we present DORAEMON , a novel cognitive-inspired framework consisting of Ventral and Dorsal Streams that mimics human navigation capabilities. The Dorsal Stream implements the Hierarchical Semantic-Spatial Fusion and Topology Map to handle spatiotemporal discontinuities, while the Ventral Stream combines CoDe-VLM and Exec-VLM to improve decision-making. Our approach also develops Nav-Ensurance to ensure navigation safety and efficiency. Extensive experimental results demonstrate the superior performance of DORAEMON.

## 6 ETHICS STATEMENT

This work adheres to the ICLR Code of Ethics. Our research did not involve human subjects or animal experimentation. All datasets used, including HM3D and MP3D, were sourced in compliance with their respective usage guidelines, ensuring no violation of privacy. We have taken care to mitigate biases in our research process, and no personally identifiable information was used.

Beyond our current study, we recognize that the responsible deployment of our framework depends on addressing broader ethical and practical challenges. A central consideration is the choice of the foundational model, which presents a trade-off between high-performance proprietary models that sacrifice transparency (e.g., Gemini) and open-source models (e.g., Qwen) that enhance accessibility and privacy but may compromise performance. Furthermore, the transition from simulation to the real world introduces serious security and reliability risks that urgently require rigorous testing. Finally, we acknowledge that inherent biases in academic datasets may limit the generalizability of such models, while the use of cameras in private spaces like homes raises fundamental privacy concerns that must be carefully managed for trustworthy adoption.

## 7 REPRODICIBILITY STATEMENT

To foster reproducibility and facilitate future research, we have made our source code and experimental setup publicly available. The anonymous repository can be accessed at:



https://anonymous.4open.science/r/DORAEMON-8D4D



This repository contains the implementation of our proposed DORAEMON framework. Additionally, we provide detailed scripts, a README.md file with environment setup instructions, and all necessary configurations to reproduce the main experimental results presented in this paper.

To further ensure that our results are reproducible, the experimental setup, including training steps and hardware details, is described comprehensively within the paper. Furthermore, all evaluations were conducted on publicly available datasets, such as HM3D and MP3D, ensuring that evaluation results can be consistently and independently verified. We believe these measures will enable other researchers to readily reproduce our work and build upon it to further advance the field.

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

## A USE OF LARGE LANGUAGE MODELS (LLMs)

Our framework centrally utilizes Large Language Models (LLMs) as the cognitive engine for its two key components: CoDe-VLM and Exec-VLM. The roles are clearly delineated: in the CoDe-VLM module, the LLM functions as a semantic compiler, translating high-level natural language instructions into a structured task knowledge graph. Subsequently, in the Exec-VLM module, the LLM acts as the executive core, performing step-by-step reasoning upon this graph to decide the optimal action.

Separate from this core function within our framework, an LLM was also utilized as a writing aid for the linguistic refinement of this manuscript. This assistance was strictly limited to improving clarity, grammar, and style. It is crucial to note that the ideation, methodology, experimental design, and data analysis were exclusively conducted by the human authors. The authors take full responsibility for all content, including the scientific claims and the final text.

## B ACTION PROPOSER

DORAEMON employs an Action Proposer(Goetting et al., 2024) to generate a refined set of candidate actions, which the Exec-VLM then evaluates for the final action decision. As shown in Figure 6, first parameterized action candidates $A_{\text{init}}^t$ are generated by the parameterized action space (Equation equation 6). Second, adaptive filtering (Equation equation 7) refines $A_{\text{cand}}^t$ using exploration state $\mathcal{V}_t$ and historical patterns $\mathcal{H}_t$. Safety-critical recovery (Equationequation 8) enforces a rotation cooldown $\gamma$ through viability evaluation $\mathcal{F}(\cdot)$. Finally, the projection module visually encodes $A_{\text{final}}^t$ into $I_{\text{anno}}^t$ with numeric tagging (0 for rotation) to interface with VLM's semantic space.

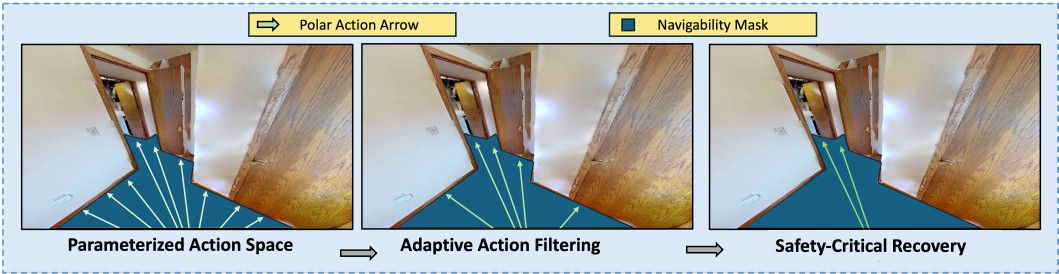

Figure 6: Action proposal: (a) Collision-free action generation within $\pm\theta_{\max}$ FOV, (b) Exploration-aware filtering with $\Delta\theta$ angular resolution, (c) Safety-constrained and action projection.

**Parameterized Action Space** Define the action space through symbolic parameters:

$$A_{\text{init}}^t = \left\{ (\theta_i, \min(\eta r_i, r_{\max})) \,\middle|\, \theta_i = k\Delta\theta, \, k \in \mathcal{K} \right\}. \tag{6}$$

where $\mathcal{K} = [-\lfloor \theta_{\max}/\Delta\theta \rfloor, \lfloor \theta_{\max}/\Delta\theta \rfloor]$ ensures full FOV coverage. The safety margin $\eta$ and collision check are derived from depth-based navigability analysis.

**Adaptive Action Filtering** Refinement combines exploration state $\mathcal{V}_t$ and historical search patterns $\mathcal{H}_t$:

$$A_{\text{cand}}^t = \left\{ (\theta_i, r_i) \in A_{\text{init}}^t \,\middle|\, \alpha(\mathcal{H}_t) \cdot s(\mathcal{V}_t) > \tau, \quad \min_{\theta_j \in A_{\text{cand}}} |\theta_i - \theta_j| \geq \theta_\delta. \right\} \tag{7}$$

where $\alpha(\cdot)$ models temporal search impact and $s(\cdot)$ quantifies spatial exploration potential.

**Safety-Critical Recovery** The next action set enforces, where $\mathcal{F}(\cdot)$ evaluates action viability and $\gamma$ controls rotation cool down:

$$A_{\text{final}}^t = \begin{cases} \{(\pi, 0)\} & \text{if }, \mathcal{F}(A_{\text{cand}}^t) \wedge (t - t_{\text{rot}} > \gamma) \\ A_{\text{cand}}^t & \text{otherwise.} \end{cases} \tag{8}$$

---

**Algorithm 1** Discrete Step Conversion

---

**Require:** Polar action $(r, \theta)$, displacement unit $\Delta_r = 0.25$m, angular unit $\Delta_\theta = 30°$

1: **if** action is `stop` **then**
2:     **return** 1                             ▷ Explicit stop handling
3: **else**
4:     $s_r \leftarrow \lceil r/\Delta_r \rceil$                   ▷ Radial step calculation
5:     $\theta_{\text{deg}} \leftarrow 180|\theta|/\pi$            ▷ Radian-degree conversion
6:     $s_\theta \leftarrow \lceil \theta_{\text{deg}}/\Delta_\theta \rceil$            ▷ Angular step calculation
7:     $N \leftarrow \max(s_r + s_\theta, 1)$         ▷ Step composition
8:     **return** $N$
9: **end if**

---

**Action Projection** The following phase focuses on visually anchoring these operational elements within the comprehensible semantic realm of the VLM. The projection component annotated visual depiction $I_{\text{anno}}^t$ from $A_{\text{final}}^t$ and $I_t$. We use numeric encoding, assigning a distinct code to each actionable option that is displayed on the visual interface. It is worth noting that rotation is assigned the code 0.

## C   STEPS CONVERSION

To establish temporal equivalence between DORAEMON's continuous actions and Habitat's discrete steps, we implement the conversion protocol formalized in Algorithm 1. Given a polar action $\mathbf{a} = (r, \theta) \in \mathbb{R}^+ \times (-\pi, \pi]$ with radial displacement $r$ meters and angular rotation $\theta$ radians:

This formulation enables direct comparison with baseline methods by normalizing both:

$$T_{\text{episode}} = \sum_{t=1}^{500} t_n \leq 500 \tag{9}$$

where $t_n$ denotes converted steps for action at time step $t$. During our experiments, one DORAEMON step $t$ was equivalent to about 9-10 $t_n$

We also presented examples of numerical conversions for steps in the experiment.

Table 5: Steps Conversion

(a) Steps Conversion for an end-to-end Step

| End-to-end Action | Non-end-to-end Steps | Non-end-to-end Action |
|---|---|---|
| $(1.27\text{m}, 53°)$ | 9 | $(1.5\text{m}, 75°)/0.25\text{m} \times 6 + 25° \times 3$ |
| $(1.7\text{m}, 60°)$ | 10 | $(1.75\text{m}, 75°)/0.25\text{m} \times 7 + 25° \times 3$ |
| $(1.1\text{m}, 93°)$ | 9 | $(1.25\text{m}, 100°)/0.25\text{m} \times 5 + 25° \times 4$ |

(b) Steps Conversion for a Navigation

| End-to-end Steps | Non-end-to-end Steps |
|---|---|
| 3 | 23 |
| 16 | 95 |
| 42 | 300 |

## D   NAVIGATION CASE

Figure 7 depicts a full navigation episode in which our memoryaugmented agent searches for a chair in an unfamiliar indoor scene. The seven consecutive frames show the agents visual observations and planned motions from entry to target acquisition. Throughout the sequence, the agent (i) reasons about semantic priorschairs are more likely near tables or in living/dining areas; (ii) fuses transient visual evidence with its episodic memory to avoid revisiting explored regions; and (iii) selects actions that maximise expected information gain while respecting safety constraints. The case illustrates how the proposed memory module complements onthefly perception to yield efficient, goaldirected exploration in cluttered, realworld layouts.

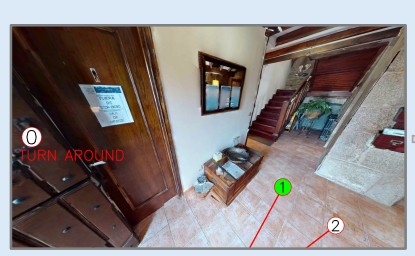

The agent is currently in a foyer where no chair is visible. Chairs are more likely deeper inside the house (e.g., living or dining rooms). Among the options, *moving forward-right takes the agent directly into the open interior* while avoiding an obstructing wall, whereas moving right is less direct and moving backward heads toward a closed door. Hence, *Action 1 (forward-right) is chosen*.

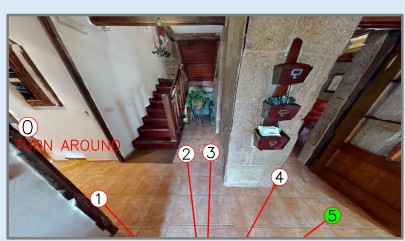

The agent is in a narrow hallway with no chair in sight. Chairs are more likely in a living space glimpsed through the partially open door to the right. Other directions head under stairs, down a narrow dead-end hall, or backward— all unlikely to reveal a chair. Therefore, *Action 5 (enter through the open door on the right) is selected*.

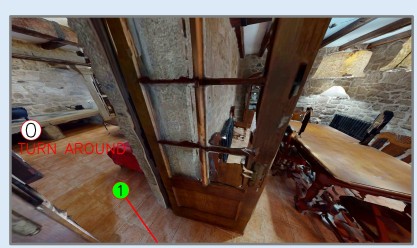

[left direction, 1.7m, 1 steps ago], […], […]

The recent memory of the hallway to my left suggests that's *an already explored area*, and going backward would be backtracking.

The agent stands in a hallway beside a damaged door. Through the opening it sees chairs in what looks like a dining room. To reach a chair quickly and avoid retracing explored paths, it *chooses Action 1 (forward-left) to pass through the doorway into that room*.

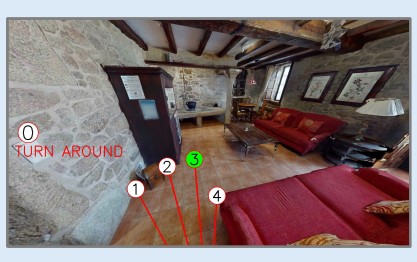

[backward direction, 1.7m, 1 steps ago], […]

memory suggests the chair might be in this room. and the area in front of me seems to be an *unexplored area*.

The agent is already in a combined living/dining room where chairs are visible, but to cover the still-unexplored central *area it chooses Action 3, which moves deepest into the room.*

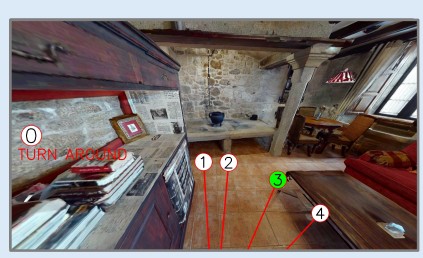

[backward direction, 1.7m, 1 steps ago], […]

*"Memory suggests the chair might be nearby, potentially in the area I just came from, but I should prioritize exploring new spaces."*

The agent is in a living/dining room and *spots chairs around a small round table on the right.* Since those visible chairs are a more reliable cue than vague earlier memories, *it chooses Action 3 to move toward that table.*

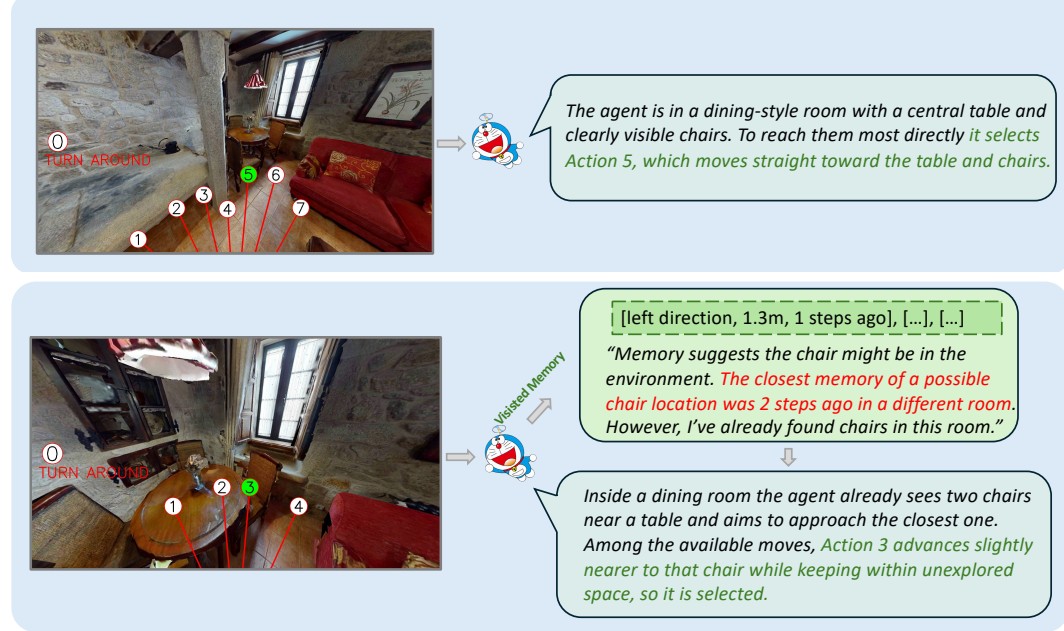

Figure 7: **Navigation case** Each row shows one decision step. *Left:* the green circle highlights the action selected for this step. *Upperright dashed green box* displays the most relevant episodic memory retrieved at this step. *Lowerright speech bubble* is the agents naturallanguage rationale that fuses (i) semantic priors, (ii) current visual evidence, and (iii) memory cues.

## E DETAILED DESCRIPTION OF AORI

### E.1 AREA OVERLAP REDUNDANCY INDEX (AORI)

The Area Overlap Redundancy Index (AORI) quantifies exploration efficiency through spatial overlap analysis. We formalize the computation with parameters from our implementation:

**Parameter Basis:**

- Map resolution: $5,000 \times 5,000$ grid (map_size=5000)
- Voxel ray casting resolution: $60 \times 60$ (voxel_ray_size=60)
- Exploration threshold: 3 observations per voxel (explore_threshold=3)
- Density scaling factor: $\eta = 0.8$ (e_i_scaling=0.8)

**Step-wise Calculation:** For each step $t \in [1, T]$:

1. Compute *observed area* $A_t = \bigcup_{i=1}^{t} \mathcal{V}(x_i, y_i)$ where $\mathcal{V}(x, y)$ is the visible region defined by:

$$\|\mathcal{V}(x, y)\| = \frac{\text{map\_size}^2}{\text{voxel\_ray\_size}^2} \cdot \pi \tag{10}$$

2. Calculate overlap ratio $r_{\text{overlap}}$:

$$r_{\text{overlap}} = \frac{\sum_{i=1}^{t-1} \mathbb{I}[\mathcal{V}(x_t, y_t) \cap \mathcal{V}(x_i, y_i) \geq \text{explore\_threshold}]}{t - 1} \tag{11}$$

3. Compute normalized density $d_{\text{normalized}}$ using Poisson expectation:

$$d_{\text{normalized}} = \min\left(1, \frac{N_{\text{obs}}}{\lambda}\right), \quad \lambda = \eta \cdot \frac{\|A_t\|}{\text{map\_size}^2} \cdot t \tag{12}$$

where $N_{\text{obs}}$ counts voxels with $\geq 3$ visits, $\lambda$ is expected active voxels

**Boundary Cases:**

- *Optimal Case* (AORI=0): When $r_{\text{overlap}} = 0$ & $d_{\text{normalized}} = 0 \Rightarrow 1 - (0.8 \cdot 1^2 + 0.2 \cdot 1) = 0$

- *Worst Case* (AORI=1):When $r_{\text{overlap}} = 1$ & $d_{\text{normalized}} = 1 \Rightarrow 1 - (0.8 \cdot 0 + 0.2 \cdot 0) = 1$

**Calculation Examples:**

- *Case1: stay still* (t=100 steps):

$$r_{\text{overlap}} = \frac{99}{99} = 1.0,$$

$$\lambda = 0.8 \cdot \frac{\pi(60/5000)^2}{1} \cdot 100 \approx 0.014,$$

$$d_{\text{norm}} = \min\left(1, \frac{100}{0.014}\right) = 1.0,$$

$$\text{AORI} = 1 - [0.8(1-1)^2 + 0.2(1-1)] = 1.0$$

(13)

- *Case2: go around* (t=500 steps):

$$r_{\text{overlap}} \approx \frac{38}{499} \approx 0.076,$$

$$\lambda = 0.8 \cdot \frac{\pi(60/5000)^2}{1} \cdot 500 \approx 0.069,$$

$$d_{\text{norm}} = \min\left(1, \frac{62}{0.069}\right) = 1.0,$$

$$\text{AORI} = 1 - [0.8 \times (1 - 0.076)^2 + 0.2 \times (1-1)] \approx 0.285$$

(14)

## F  EXPERIMENTAL SETUP DETAILS

**Implementation Details.** The maximal navigation steps per episode are set to 40. The agent's body has a radius of $0.17m$ and a height of $1.5m$. Its RGB-D sensors are positioned at $1.5m$ height with a $-0.45$ radian downward tilt and provide a $131°$ Field of View (FoV). For rotation, the agent selects an angular displacement corresponding to one of 60 discrete bins that uniformly discretize the $360°$ range. Success requires stopping within $d_{\text{success}} = 0.3$m of the target object and visually confirming it. Success requires stopping within $d_{\text{success}} = 0.3m$ of the target object and visually confirming it. Our DORAEMON framework primarily utilizes `Gemini-1.5-pro` as the VLM and `CLIP ViT-B/32` for semantic embeddings, with caching implemented for efficiency. Key hyperparameters include: topological map connection distance $\delta_{\text{connect}} = 1.0$m, node update interval $S_{\text{update}} = 3$ steps, $L_1$ hierarchical clustering weight $w = 0.4$, AORI grid resolution $\delta_{\text{grid}} = 0.1$m, minimum obstacle clearance $d_{\text{min\_obs}} = 0.5$m, and various stuck detection thresholds (e.g., path inefficiency $\eta_{\text{path}} < 0.25$, small area coverage $\delta_{\text{area\_gain}} < 0.35\text{m}^2$, high rotation/translation ratio $\rho_{\text{rot/trans}} > 2.0$ for short paths when $\|\text{path}\| < 0.5$m) and a precision movement factor $\gamma_{\text{step}} = 0.1$.

## G  HIERARCHICAL CONSTRUCTION

### G.1  LEVEL $L_3$: OBSERVATION ANCHORING

- **Input**: Raw topological nodes $v_t \in \mathcal{V}$ from Eq 1

- **Process**: Directly mapping to memory nodes

$$h_j^{(3)} = \left(\text{id}_j^{(3)}, L_3, \emptyset, \{v_t\}\right).$$

(15)

- **Output**: $h_j^{(3)}$ nodes storing original $p_t, \mathbf{s}_t$ from $v_t$

## G.2 LEVEL $L_2$: AREA FORMATION ($L_3 \rightarrow L_2$)

- **Input**: $h_j^{(3)}$ nodes with spatial coordinates $p_t$
- **Clustering**:
    1. Compute combined distance:

    $$d_{\text{comb}} = 0.4\|p_i - p_j\|_2 + 0.6\left(1 - \frac{\mathbf{s}_i \cdot \mathbf{s}_j}{\|\mathbf{s}_i\|\|\mathbf{s}_j\|}\right). \qquad (16)$$

    2. Apply adaptive threshold:

    $$\theta_1' = \begin{cases} 1.5\theta_1 & (|\mathcal{O}| > 20) \\ 0.8\theta_1 & (|\mathcal{O}| < 10) \\ \theta_1 & \text{otherwise.} \end{cases} \qquad (17)$$

    3. Generate clusters using scipy.linkage + fcluster
- **Functional Labeling**:

$$\texttt{area\_type} = \arg\max_t \sum_{v \in \mathcal{C}_j^{(2)}} \sum_{k \in K_t} \mathbb{I}[k \in v.L_t]. \qquad (18)$$

- **Output**: $h_m^{(2)}$ nodes with:
    - Parent: $h_n^{(1)}$ ($L_1$ room node).
    - Children: $\{h_j^{(3)}\}$ ( observations).
    - Spatial boundary: Convex hull of $p_t$ positions.

## G.3 LEVEL $L_1$: ROOM FORMATION ($L_2 \rightarrow L_1$)

- **Input**: $h_m^{(2)}$ areas with spatial centroids $P_A$
- **Two-stage Clustering**:
    1. *Spatial Pre-clustering*:

    $$C_{\text{spatial}} = \text{fcluster}(\text{linkage}(d_{\text{spatial}}), \theta_2 = 3.0\text{m}). \qquad (19)$$

    2. *Functional Refinement*:

    $$\mathcal{F}_s = \{\mathcal{A}_{s,f} | f = \text{MapToRoomFunction}(\texttt{area\_type})\}. \qquad (20)$$

- **Output**: $h_n^{(1)}$ nodes containing:
    - Parent: $h_0^{(0)}$ ($L_0$ root)
    - Children: $\{h_m^{(2)}\}$ ($L_2$ areas)

## G.4 LEVEL $L_0$: ENVIRONMENT ROOT

- **Input**: All $h_n^{(1)}$ room nodes
- **Consolidation**:

$$h_0^{(0)} = \left(\texttt{GLOBAL\_ROOT}, L_0, \emptyset, \{h_n^{(1)}\}\right). \qquad (21)$$

- **Function**: Global access point for memory queries

# H MEMORY RETRIEVAL SCORING DETAILS

## H.1 SCORING FUNCTION DECOMPOSITION

The retrieval score combines four evidence components through weighted summation:

$$S(h_i) = 0.45 S_{\text{sem}} + 0.30 S_{\text{spa}} + 0.20 S_{\text{key}} + 0.05 S_{\text{time}}. \qquad (22)$$

## H.2 COMPONENT SPECIFICATIONS

### H.2.1 SEMANTIC SIMILARITY

- **Input**: CLIP embeddings $\mathbf{s}_q$ (query) and $\mathbf{s}_i$ (node)
- **Calculation**:

$$S_{\text{sem}} = \frac{1}{2}\left(1 + \frac{\mathbf{s}_q^\top \mathbf{s}_i}{\|\mathbf{s}_q\|\|\mathbf{s}_i\|}\right) \in [0, 1]. \tag{23}$$

### H.2.2 SPATIAL PROXIMITY

- **Input**: Agent position $p_a$, node position $p_i$
- **Decay function**:

$$S_{\text{spa}} = \exp\left(-\frac{\|p_a - p_i\|_2}{5.0}\right). \tag{24}$$

### H.2.3 KEYWORD RELEVANCE

- **Input**: Query terms $T$, node keywords $K_i$ (from $L_t$)
- **Matching score**:

$$S_{\text{key}} = \frac{|T \cap K_i|}{\max(|T|, 1)}. \tag{25}$$

### H.2.4 TEMPORAL RECENCY

- **Input**: Current time $t_c$, observation time $t_i$
- **Decay model**:

$$S_{\text{time}} = \exp\left(-\frac{|t_c - t_i|}{600}\right). \tag{26}$$

## H.3 PARAMETER CONFIGURATION

Table 6: Scoring Component Weights

| Component | Symbol | Value |
|---|---|---|
| Semantic Similarity | $\alpha_{\text{sem}}$ | 0.45 |
| Spatial Proximity | $\alpha_{\text{spa}}$ | 0.30 |
| Keyword Relevance | $\alpha_{\text{key}}$ | 0.20 |
| Temporal Recency | $\alpha_{\text{time}}$ | 0.05 |

## H.4 SEARCH PROCESS

The beam search executes through these discrete phases:

**Initialization Phase**

- Start from root node(s): $\mathcal{F}_0 = \{h_{\text{root}}\}$
- Set beam width: $B = 5$

**Iterative Expansion** For each hierarchy level $l \in \{L_3, L_2, L_1, L_0\}$:

- Score all children: $S(h_{\text{child}}) \forall h_{\text{child}} \in \mathcal{C}(h_j), h_j \in \mathcal{F}_l$
- Select top-$B$ nodes

**Termination Conditions**

- **Success**: Reached $L_0$ nodes and selected top-$K$ results
- **Failure**: No nodes satisfy $S(h_i) > 0.4$ threshold

## H.5 COMPUTATIONAL PROPERTIES

- **Time Complexity**: $O(B \cdot D)$ for depth $D = 4$
- **Memory Complexity**: $O(B)$ nodes per level
- **Score Normalization**:

$$\sum_{k \in \{\text{sem,spa,key,time}\}} \alpha_k = 1.0. \tag{27}$$

# I CHAIN-OF-THOUGHT PROMPT

Our Exec-VLM leverages a structured Chain-of-Thought (CoT) prompt to guide the decision-making process. The complete prompt is provided below:

```
TASK: NAVIGATE TO THE NEAREST [TARGET_OBJECT], and get as close to it as
    ↪ possible.
Use your prior knowledge about where items are typically located within
    ↪ a home.
There are [N] red arrows superimposed onto your observation, which
    ↪ represent potential actions.
These are labeled with a number in a white circle, which represent the
    ↪ location you would move to if you took that action.
[TURN_INSTRUCTION]

Let's solve this navigation task step by step:

1. Current State Analysis: What do you observe in the environment? What
    ↪ objects and pathways are visible?
   Look carefully for the target object, even if it's partially visible
       ↪ or at a distance.

2. Memory Integration: Review the memory context below for clues about
    ↪ target location.
   - Pay special attention to memories containing or near the target
       ↪ object
   - Use recent memories (fewer steps ago) over older ones
   - Consider action recommendations based on memory

3. Goal Analysis: Based on the target and home layout knowledge, where
    ↪ is the [TARGET_OBJECT] likely to be?

4. Scene Assessment: Quickly evaluate if [TARGET_OBJECT] could
    ↪ reasonably exist in this type of space:
   - If you're in an obviously incompatible room (e.g., looking for a
       ↪ [TARGET_OBJECT] but in a clearly different room type), choose
       ↪ action 0 to TURN AROUND immediately

5. Path Planning: What's the most promising direction to reach the
    ↪ target? Avoid revisiting
   previously explored areas unless necessary. Consider:
   - Available paths and typical room layouts
   - Areas you haven't explored yet

6. Action Decision: Which numbered arrow best serves your plan? Return
    ↪ your choice as {"action": <action_key>}. Note:
   - You CANNOT GO THROUGH CLOSED DOORS, It doesn't make any sense to go
       ↪ near a closed door.
   - You CANNOT GO THROUGH WINDOWS AND MIRRORS
```

```
     - You DO NOT NEED TO GO UP OR DOWN STAIRS
     - Please try to avoid actions that will lead you to a dead end to
         ↪ avoid affecting subsequent actions, unless the dead end is very
         ↪ close to the [TARGET_OBJECT]
     - If you see the target object, even partially, choose the action
         ↪ that gets you closest to it
```

## J DETAILED DESCRIPTION OF BASELINE

To assess the performance of *DORAEMON*, we compare it with **16** recent baselines for (zeroshot) objectgoal navigation. Summaries are given below.

**ProcTHOR** (Deitke et al., 2022): A procedurallygenerated 10Kscene suite for largescale Embodied AI.

**ProcTHOR_ZS** (Deitke et al., 2022): ProcTHOR_ZS trains in ProcTHOR and evaluates zeroshot on unseen iTHOR/RoboTHOR scenes to test crossdomain generalisation.

**SemEXP** (Chaplot et al., 2020): Builds an online semantic map and uses goaloriented exploration to locate the target object efficiently, achieving stateoftheart results in Habitat ObjectNav 2020.

**HabitatWeb** (Ramrakhya et al., 2022): Collects largescale human demonstrations via a browser interface and leverages behaviour cloning to learn objectsearch strategies.

**PONI** (Ramakrishnan et al., 2022): Learns a potentialfield predictor from static supervision, enabling interactionfree training while preserving high navigation success.

**ZSON** (Majumdar et al., 2022): Encodes multimodal goal embeddings (text + images) to achieve zeroshot navigation towards previously unseen object categories.

**PSL** (Sun et al., 2024): Prioritised Semantic Learning selects informative targets during training and uses semantic expansion at inference for zeroshot instance navigation.

**PixelNav** (Cai et al., 2024): Introduces pixelguided navigation skills that bridge foundation models and ObjectNav, relying solely on RGB inputs.

**SGM** (Zhang et al., 2024b): Imagine Before Go constructs a selfsupervised generative map to predict unseen areas and improve exploration efficiency.

**ImagineNav** (Zhao et al., 2024): Prompts visionlanguage models to imagine future observations, guiding the agent toward informationrich viewpoints.

**CoW** (Gadre et al., 2023): Establishes the Cows on Pasture benchmark for languagedriven zeroshot ObjectNav and releases baseline policies without indomain training.

**ESC** (Zhou et al., 2023): Employs soft commonsense constraints derived from language models to bias exploration, markedly improving zeroshot success over CoW.

**L3MVN** (Yu et al., 2023): Utilises large language models to reason about likely room sequences, while a visual policy executes the suggested path.

**VLFM** (Yokoyama et al., 2024): Combines VLM goallocalisation with frontierbased exploration, removing the need for reinforcement learning or taskspecific finetuning.

**VoroNav** (Wu et al., 2024): Simplifies the search space via Voronoi partitions and pairs this with LLMdriven semantic planning for improved zeroshot performance.

**TopVNav** (Zhong et al., 2024): Lets a multimodal LLM perform spatial reasoning directly on topview maps, with adaptive visual prompts for globallocal coordination.

**SGNav** (Yin et al., 2024): Online builds a 3D scene graph and uses hierarchical Chain-of-Thought prompting so an LLM can infer probable target locations.

