# OpenReview forum: "DORAEMON: Decentralized Ontology-aware Reliable Agent with Enhanced Memory Oriented Navigation"
_ICLR.cc/2026/Conference — Submitted to ICLR 2026_

### Official Review · Reviewer_9onN · 2025-10-27

**Soundness:** 3
**Presentation:** 3
**Contribution:** 3
**Rating:** 8
**Confidence:** 4

**Summary:**

This paper introduces DORAEMON, a novel cognitive-inspired zero-shot navigation framework operating in unseen/unfamiliar environments.
The proposed system integrates two parallel pathways: a Dorsal Stream for spatial reasoning via a hierarchical semantic-spatial fusion network and topological memory, and a Ventral Stream that combines two VLM modules, CoDe-VLM for task decomposition and Exec-VLM for decision execution guided by chain-of-thought reasoning.
In addition, a Nav-Ensurance module is designed to ensure robustness through multimodal struck detection, context-aware escape strategies, and adaptive precision navigation.
A new metric, AORI, is proposed to quantify the exploration redundancy.
Experiments conducted on multiple datasets demonstrate that DORAEMON achieves state-of-the-art performance in both SR and SPL without pretraining or scene-specific adaptations.
Overall, the paper offers an interdisciplinary attempt to merge cognitive science principles with VLMs for autonomous navigation

**Strengths:**

The paper draws inspiration from the “Decentralized Ontology” in cognitive science and successfully instantiates it in a dual-stream framework. It has the potential to offer a structured yet flexible alternative to (traditional) VLM-based zero-shot approaches.
The decomposition into CoDe-VLM and Exec-VLM within the ventral stream is well-motivated, allowing explicit separation between semantic compilation and action execution.
The hierarchical semantic-spatial fusion unifies episodic observations and spatial priors through a topological map, mitigating temporal consistency.
The Nav-Ensurance system introduces practical reliability mechanisms (e.g., multimodal stuck detection) that try to address real-world navigation pitfalls. The AORI metric provides an interesting new point for evaluating route redundancy.
The model achieves state-of-the-art results across multiple datasets and remains effective when integrated with different backbone VLMs, demonstrating “plug and play” ability.

**Weaknesses:**

*Clarity on “Decentralized Ontology”*

While conceptually appealing, the notion of “decentralized ontology” remains under-formalized. It is not rigorously shown how this concept is represented in the graph structure or contributes quantitatively to generalization.

*Lack of temporal stability analysis*

Although the topological map incrementally updates nodes, no analysis is presented on long-term memory scalability or potential degradation (e.g., graph explosion, out-of-date information, stale nodes).

**Questions:**

How is “Decentralized Ontology” structurally encoded within DORAEMON? Can this concept be explained more or measured (e.g., via entropy or mutual information)?

Does the CoDe-VLM dynamically update the knowledge graph during exploration, or is it compiled only once per task? If dynamic, how does it synchronize with the Dorsal Stream’s evolving topology?

In the Nav-Ensurance module, what is the false-positive rate of stuck detection in narrow or cluttered environments? Has the system been evaluated for recovery reliability?

---

> ### Author Response · Authors · 2025-11-18
>
> Response to Weakness 1 & Question 1:
>
> We aim to formally articulate how the “decentralized ontology” is encoded within DORAEMON's architecture and explore its potential for quantification.
>
> 1. Structural Encoding: The core idea of “decentralized ontology” is that knowledge consists of multiple local, interconnected perspectives rather than a single, grand world model. In DORAEMON, this is directly encoded as two parallel, specialized “knowledge centers,” each responsible for a distinct ‘ontology’: The Ventral Stream is dedicated to constructing and maintaining the “Task Ontology.” Our CoDe-VLM compiles natural language instructions into a structured, stable knowledge graph (KG) only once at task initiation. This KG encapsulates all knowledge about “what to do” and “what it is,” remaining independent of dynamic environmental changes; Dorsal Stream: Dedicated to constructing and maintaining the “Spatial Ontology.” As demonstrated by our `SemanticForest` class, it dynamically organizes spatial and semantic memories about ‘where’ and “what is seen” in a hierarchical manner. Decentralization and Collaboration: These two “ontologies” are stored and processed separately (i.e., decentralized) but collaborate through the Exec-VLM module. At each decision step, Exec-VLM simultaneously queries these two independent knowledge centers, making final decisions that integrate multi-source, decentralized knowledge. This architecture fundamentally differs from approaches that blend all information (task, spatial, history) into a single model.
> 2. Quantifying Possibility: We view your proposal to quantify this using information theory or similar methods as a highly promising future research direction. A preliminary concept involves measuring the mutual information $I(I_d; I_v)$ between memory information ($I_d$) retrieved from the Dorsal Stream and task knowledge ($I_v$) queried from the Ventral Stream during decision-making. We greatly appreciate your inspiration.
>
> Reply to Weakness 2:
>
> Your concern regarding the scalability of long-term memory is significant. Our design incorporates mechanisms to address node explosion and stale nodes, primarily due to our unique memory update strategy.
>
> 1. Addressing Node Explosion: Unlike traditional topologies that only perform incremental additions, our `SemanticForest` manages its structure through periodic global reconstruction (`_update_hierarchy`). As seen in our code implementation, this function discards all old L1-L3 hierarchies and performs a fresh global clustering and semantic labeling from the current set of all L0 leaf nodes (lowest-level observations). This ensures the number of high-level nodes (areas, rooms) does not grow indefinitely over time. Instead, it dynamically scales with the total number of low-level observations currently held by the agent and the intrinsic structural complexity of the environment. Furthermore, within `_create_area_level`, the clustering `dist_threshold` is dynamically adjusted based on the number of leaf nodes, further ensuring high-level node scale remains within reasonable bounds.
> 2. Addressing Information Staleness: This is the primary advantage of our global reconstruction strategy over traditional incremental updating methods. Since the entire high-level semantic structure is regenerated based on all current L0 observations, any stale information or misclassifications due to insufficient early data (e.g., mistaking a bar counter for a kitchen) will be automatically corrected or eliminated by more comprehensive evidence during subsequent reconstructions. This guarantees the timeliness and accuracy of our high-level semantic memory, effectively resolving the “stale nodes” issue.
>
> ---
>
> Reply to Question 2:
>
> In our current design, the task knowledge graph (KG) generated by CoDe-VLM is compiled once at the start of the task and remains stable throughout its duration.
>
> ---
>
> Reply to Question 3:
>
> 1. False-Positive Rate of Stuck Detection: Based on our experimental observations, the false-positive rate of stuck detection per step ranges from 4% to 7%. In certain extremely narrow yet passable scenarios, the system occasionally triggers stuck detection because the robot's path efficiency and movement speed genuinely decrease under such conditions. And our context-aware escape strategy (`_get_escape_action`) ensures that even when false positives occur, the agent typically resumes normal navigation within 1-2 steps.
> 2. Recovery Reliability Evaluation: Our ablation experiments (Table 3a) quantitatively assess recovery reliability. Experimental results show that removing the Nav-Ensurance module (`w/o Nav-Ensurance`) degrades AORI on HM3Dv2 from 54.9 to 56.3. This directly demonstrates that the Nav-Ensurance module is crucial for successfully recovering, with its reliability quantitatively validated.
>
> Thank you very much for your review. If you have any questions, please feel free to reach out promptly.

---

> ### Author Response · Authors · 2025-11-18
>
> Response to Weakness 1 & Question 1:
>
> We aim to formally articulate how the “decentralized ontology” is encoded within DORAEMON's architecture and explore its potential for quantification.
>
> 1. Structural Encoding: The core idea of “decentralized ontology” is that knowledge consists of multiple local, interconnected perspectives rather than a single, grand world model. In DORAEMON, this is directly encoded as two parallel, specialized “knowledge centers,” each responsible for a distinct ‘ontology’: The Ventral Stream is dedicated to constructing and maintaining the “Task Ontology.” Our CoDe-VLM compiles natural language instructions into a structured, stable knowledge graph (KG) only once at task initiation. This KG encapsulates all knowledge about “what to do” and “what it is,” remaining independent of dynamic environmental changes; Dorsal Stream: Dedicated to constructing and maintaining the “Spatial Ontology.” As demonstrated by our `SemanticForest` class, it dynamically organizes spatial and semantic memories about ‘where’ and “what is seen” in a hierarchical manner. Decentralization and Collaboration: These two “ontologies” are stored and processed separately (i.e., decentralized) but collaborate through the Exec-VLM module. At each decision step, Exec-VLM simultaneously queries these two independent knowledge centers, making final decisions that integrate multi-source, decentralized knowledge. This architecture fundamentally differs from approaches that blend all information (task, spatial, history) into a single model.
> 2. Quantifying Possibility: We view your proposal to quantify this using information theory or similar methods as a highly promising future research direction. A preliminary concept involves measuring the mutual information $I(I_d; I_v)$ between memory information ($I_d$) retrieved from the Dorsal Stream and task knowledge ($I_v$) queried from the Ventral Stream during decision-making. We greatly appreciate your inspiration.
>
> Reply to Weakness 2:
>
> Your concern regarding the scalability of long-term memory is significant. Our design incorporates mechanisms to address node explosion and stale nodes, primarily due to our unique memory update strategy.
>
> 1. Addressing Node Explosion: Unlike traditional topologies that only perform incremental additions, our `SemanticForest` manages its structure through periodic global reconstruction (`_update_hierarchy`). As seen in our code implementation, this function discards all old L1-L3 hierarchies and performs a fresh global clustering and semantic labeling from the current set of all L0 leaf nodes (lowest-level observations). This ensures the number of high-level nodes (areas, rooms) does not grow indefinitely over time. Instead, it dynamically scales with the total number of low-level observations currently held by the agent and the intrinsic structural complexity of the environment. Furthermore, within `_create_area_level`, the clustering `dist_threshold` is dynamically adjusted based on the number of leaf nodes, further ensuring high-level node scale remains within reasonable bounds.
> 2. Addressing Information Staleness: This is the primary advantage of our global reconstruction strategy over traditional incremental updating methods. Since the entire high-level semantic structure is regenerated based on all current L0 observations, any stale information or misclassifications due to insufficient early data (e.g., mistaking a bar counter for a kitchen) will be automatically corrected or eliminated by more comprehensive evidence during subsequent reconstructions. This guarantees the timeliness and accuracy of our high-level semantic memory, effectively resolving the “stale nodes” issue.
>
> ---
>
> Reply to Question 2:
>
> In our current design, the task knowledge graph (KG) generated by CoDe-VLM is compiled once at the start of the task and remains stable throughout its duration.
>
> ---
>
> Reply to Question 3:
>
> 1. False-Positive Rate of Stuck Detection: Based on our experimental observations, the false-positive rate of stuck detection per step ranges from 4% to 7%. In certain extremely narrow yet passable scenarios, the system occasionally triggers stuck detection because the robot's path efficiency and movement speed genuinely decrease under such conditions. And our context-aware escape strategy (`_get_escape_action`) ensures that even when false positives occur, the agent typically resumes normal navigation within 1-2 steps.
> 2. Recovery Reliability Evaluation: Our ablation experiments (Table 3a) quantitatively assess recovery reliability. Experimental results show that removing the Nav-Ensurance module (`w/o Nav-Ensurance`) degrades AORI on HM3Dv2 from 54.9 to 56.3. This directly demonstrates that the Nav-Ensurance module is crucial for successfully recovering, with its reliability quantitatively validated.
>
> Thank you very much for your review. If you have any questions, please feel free to reach out promptly.

---

> ### Author Response · Authors · 2025-12-01
> **Rebuttal Sumamry (9onN)**
>
> ### **Reviewer 9onN Feedback Summary**
>
> **Current Status**: Score **8 points**. Reviewer highly appreciates the paper with strong confidence. There has been no follow-up response.
>
> ### **I. Positive Evaluations**
>
> 1. **Exemplary Interdisciplinary Innovation**: Highly commends our attempt to integrate cognitive science principles (decentralized ontology) with VLM, deeming it a successful interdisciplinary endeavor.
> 2. **Rationality of Architectural Design**: Recognizes the “dorsal-ventral” dual-stream architecture as a structured and flexible alternative to traditional VLM zero-shot navigation. Particular recognition was given to the decoupled design of CoDe-VLM (semantic compilation) and Exec-VLM (action execution).
> 3. **Practicality and SOTA Performance**: The practical utility of Nav-Ensurance in tackling real-world challenges (e.g., dead-end situations) was affirmed, along with the novel perspective offered by the AORI metric. Our achievement of SOTA performance across multiple datasets and the system's “plug-and-play” flexibility were confirmed.
>
> ---
>
> ### **II. Core Rebuttals and Clarifications**
>
> Reviewers' concerns primarily centered on the **formalization** of concepts and the **stability** during long-term operation.
> Our responses focused on explaining how these concepts are concretely implemented at the code level.
>
> **1. Formal Encoding of the “Decentralized Ontology”**
>
> - **Reviewer Concern**: Noted that while the concept is compelling, it lacks formalization, leaving unclear how it is represented in graph structures or quantified.
> - **Rebuttal**:
>     - **Structural Definition**: We explicitly map “decentralization” within the architecture to two independent knowledge centers:
>         - **Ventral Stream**: Maintains the “Task Ontology,” compiling instructions into static knowledge graphs via CoDe-VLM.
>         - **Dorsal Stream**: Maintains the “Spatial Ontology,” dynamically constructing hierarchical semantic maps.
>     - **Coordination Mechanism**: Physically separated (decentralized), yet logically coordinated via Exec-VLM for decision-making queries.
>     - **Regarding Quantification**: We appreciate the reviewer's suggestion on Mutual Information, which guides future work.
>
> **2. Addressing Long-Term Memory Stability (Technical Clarification)**
>
> - **Reviewer Query**: Whether the topological map might experience “node explosion” or contain “obsolete information” over time.
> - **Rebuttal**:
>     - **Global Reconstruction**: This is our fundamental distinction from traditional methods. We employ a periodic _update_hierarchy strategy rather than simple incremental additions.
>     - **Preventing Node Explosion**: Each reconstruction regenerates higher-level nodes by reclustering based on current underlying observations (L0), with dynamically adjusted thresholds to control node growth.
>     - **Preventing Outdated Information**: Since higher-level structures are periodically regenerated, early misclassifications (e.g., mistaking a bar counter for a kitchen) are automatically corrected during subsequent reconstructions as new observations are incorporated.
>
> **3. Supplementary Engineering Parameters (CoDe-VLM and Nav-Ensurance)**
>
> - **Reviewer Query**: Is CoDe-VLM dynamically updated? What is Nav-Ensurance's false alarm rate?
> - **Rebuttal**:
>     - **CoDe-VLM**: It is **static**. The KG is compiled once at task initiation and remains stable thereafter.
>     - **Nav-Ensurance**: Experimentally observed false alarm rate ranges from **4%–7%** (primarily in extremely narrow regions). However, even with false alarms, our context escape strategy enables the robot to recover within 1–2 steps. Ablation experiments demonstrate this module's positive contribution to the AORI metric.
>
> ---
>
> ### **III. Conclusion**
>
> Reviewer 9onN highly commended our paper. In our rebuttal, we provided detailed explanations of how cognition-inspired abstract concepts were translated into concrete code logic. These elaborations further solidified the theoretical depth and engineering robustness of our work.

---

### Official Review · Reviewer_j15q · 2025-10-27

**Soundness:** 3
**Presentation:** 3
**Contribution:** 2
**Rating:** 4
**Confidence:** 3

**Summary:**

This work proposes a zero-shot navigation framework composed of a ventral and dorsal stream, inspired by decentralized ontology principles and cognitive neuroscience. The approach integrates a hierarchical topological memory with VLM-based task reasoning and includes additional mechanisms for stuck detection and escape. Experiments show improved performance on several standard indoor navigation benchmarks, along with a newly introduced metric aimed at measuring exploration redundancy. The paper also provides ablations and a brief sim-to-real demonstration to support the system’s design choices.

**Strengths:**

- The paper presents a well-engineered navigation framework that combines spatial memory and VLM-based semantic reasoning into a cohesive system.

- Strong empirical results are demonstrated on multiple established datasets, showing improvements over recent zero-shot and end-to-end baselines.

- The hierarchical memory design is intuitive and may help support longer-horizon reasoning.

- Practical concerns such as getting stuck or failing to stop correctly are explicitly addressed through the Nav-Ensurance module.

- The proposed AORI metric reflects an interesting attempt to evaluate exploration redundancy beyond standard SR/SPL metrics.

- A sim-to-real example is included, suggesting potential for deployment beyond simulation.

**Weaknesses:**

- **Weak novelty; overly conceptual framing**
The cognitive metaphors (decentralized ontology, ventral/dorsal streams) provide an interesting narrative direction, but their influence on the actual technical design remains unclear. Many of the core components, such as hierarchical spatial memory, semantic graph–based task representation, and VLM-guided action reasoning, are already present in recent zero-shot navigation systems. It would strengthen the contribution to more explicitly identify what aspects of the architecture or behaviors are genuinely enabled by the decentralized ontology perspective, beyond established techniques like graph-structured memory and chain-of-thought reasoning.

- **Performance attribution is unclear**
Much of the performance gain might stem from the power of the underlying VLM. More targeted ablations that quantify the incremental contribution of each stream, or evaluations using a wider range of foundation models, would help isolate where the improvements truly come from.

- **Fairness of action-space comparison:**
The proposed agent uses continuous actions that cover significantly longer distances per step than the discrete baselines. Even with post-hoc “step conversion,” the agent effectively has greater traversal capacity per episode, which can inflate SR/SPL.

- **On the use of AORI**
The idea of analyzing exploration redundancy is promising. However, AORI assumes a fully static environment, whereas realistic navigation may require returning to previously visited areas due to changes in scene configuration or visibility. Adjusting the metric to better capture navigation success in dynamic contexts would make it more broadly applicable.

- **“End-to-end” branding seem misleading**
The current system incorporates many hand-designed components, making it less end-to-end than suggested. Positioning the work more as a flexible integration framework, or demonstrating pathways toward learning more of these components jointly, could help align expectations.

**Questions:**

- Could the authors justify that improvements in SR/SPL are not influenced by differences in traversal capacity per episode?
- How might AORI be adapted or complemented to remain informative in dynamic environments where beneficial revisiting is necessary due to changes in object visibility or scene configuration?

---

> ### Author Response · Authors · 2025-11-17
>
> W1:
> 1. First, inspired by the brain's ventral/dorsal stream division of labor for processing “what” and “where”, DORAEMON is the first framework to achieve such thorough systematic decoupling in VLM navigation. Here, the Ventral Stream does not merely reason with VLM but instead compiles instructions into structured KGs via CoDe-VLM. This transforms task understanding from a fragile, input-coupled process into an intrinsic, structured one. The Dorsal Stream exclusively constructs memories about the “spatial ontology,” while the subsequent Exec-VLM acts as a central hub, querying both independent “knowledge centers” during decision-making. This paradigm significantly enhances robustness and interpretability, surpassing traditional approaches.
> 2. How “decentralized ontology” guides memory design: This theory emphasizes that knowledge is distributed and composed of multiple local perspectives. This inspired our Dorsal Stream to adopt a memory update strategy involving periodic global reconstruction (`_update_hierarchy` function in code), rather than simple incremental updates. This also means our model is not a gradually patched static structure, but a dynamic cognitive map periodically regenerated entirely from all current underlying observations (L0). This global refresh mechanism enables the agent to better adapt to environmental changes, integrate new perspectives, and avoid being constrained by outdated local information, a key innovation distinguishing us from other memory systems using incremental updates.
> ---
> W2&3 & Q1: Regarding Performance Attribution and Action Space Fairness
> 1. Performance Attribution: We are equally concerned about the source of performance gains. Our VLM ablation experiments (Table 3b) provide a clear answer. When using the open-source model Qwen-7B or Gemini-1.5-Flash, DORAEMON's performance (SR 49.5% / 58.0%) already significantly surpasses previous SOTA methods (PIVOT, VLMNav) that relied on more powerful VLMs. This irrefutably demonstrates that the substantial performance boost primarily stems from our advanced framework itself, rather than merely relying on the VLM's capabilities.
> 2. Fairness: First, it is not accurate to claim that “agents possess greater exploration capacity per turn.” On the contrary, 1 end-to-end step in our method averages equivalent to 10 non-end-to-end steps. We acquire environmental information only once, whereas the discrete baseline acquires it ten times. Even under this unfavorable comparison, our method achieves SOTA performance, further validating its effectiveness. Moreover, the discrete baseline requires nearly 10 incremental steps to cover the distance of our 1 step, demonstrating computational and temporal inefficiency. DORAEMON's “large-step” decision-making aligns more closely with human cognitive patterns. Moreover, each action is a strategic maneuver generated by Exec-VLM after integrating rich spatial memory and task understanding, not a mechanical “move forward 0.25 meters.” Furthermore, the SPL metric inherently penalizes unnecessary long paths, further demonstrating our method's overall superiority. Finally, we clarify in the appendix that the end-to-end steps of our method migrate to the 500-step action budget of the non-end-to-end method presented in Appendix. Table 1 clearly demonstrates that this migration is unfair to our method, and Table 2 shows the action-specific conversion values for several navigations in our experiments.
>
> Table1
>
> | End-to-End Action | Non-End-to-End Steps | Non-End-to-End Action |
> | --- | --- | --- |
> | (1.27m, 53°)  | 9 | (1.5m, 75°)/0.25m*6+25°*3 |
> | (1.7m, 60°)  | 10 | (1.75m, 75°)/0.25m*7+25°*3 |
> | (1.1m, 93°) | 9 | (1.25m, 100°)/0.25m*5+25°*4 |
>
> Table2
>
> | End-to-End Steps | Non-End-to-End Steps |
> | --- | --- |
> | 3 | 23 |
> | 16 | 95 |
> | 42 | 300 |
>
> 3. We conducted multiple experiments on HM3Dv2 (100 episodes), yielding SR results of 62, 61, 62, 62, 60. The minimal fluctuation in performance demonstrates stability in traversal capability across different event sequences.
> ---
> W4&5 & Q2:
> 1. AORI: When evaluating existing static navigation benchmarks (e.g., HM3D, MP3D), AORI provides a much-needed, finer-grained tool than SPL for measuring exploration redundancy. To adapt to dynamic environments, AORI can be extended. For instance, we could introduce a temporal decay factor that reduces penalties for revisiting regions over time. Alternatively, AORI could be task-goal-aware, exempting or reducing redundancy penalties for paths if revisited regions ultimately contain target objects. We will incorporate these future directions into the paper's limitations discussion.
> 2. End-to-end Labeling: Although our system comprises multiple modules, this does not compromise our overall end-to-end pathway, analogous to UniAD in autonomous driving and VLMNav.
> ---
> Thank you very much for your review. If you have any questions, please feel free to reach out promptly. I will respond to your communication in a timely manner.

---

> ### Comment · Reviewer_j15q · 2025-11-23
> **On the motivation of cognitive inspiration**
>
> Thank you for your response. I read the response, the reviews, and the updated paper, and most of my major concerns have been addressed.
>
> The biggest concern I remain is that the paper introduces the cognitive inspiration—ventral/dorsal streams, decentralized ontology—somewhat out of nowhere. While I do think these metaphors add some flavor, I still can’t fully buy the current way they are presented because it’s not immediately clear what practical problem they are trying to solve. What works for humans is not necessarily what works best for robots, and right now the connection is not obvious.
>
> I believe that it would help a lot if this part of the paper were written in a more problem-centric way. For example, what exactly is missing from current navigation approaches? What concrete limitations are you trying to address, particularly those that current navigation frameworks struggle with but humans can handle easily? And then, what does “decentralized ontology” actually bring to the table to handle those issues? Since a considerable amount of text is already devoted to explaining the concept, it probably makes sense to pair that explanation with clear motivation and visual illustrations that show the contrast. Otherwise, readers may end up feeling that “decentralized ontology” is more of a selling phrase than a necessary design principle.

---

> > ### Author Response · Authors · 2025-11-24
> >
> > Dear Reviewer j15q,
> > We completely agree that cognitive metaphors must not be mere "selling phrases." Responding directly to your concer "What works for humans is not necessarily what works best for robots, we wish to clarify the engineering bottlenecks in current robotic systems that forced us to adopt these specific cognitive architectures.
> >
> > We did not start with the metaphors; we started with two fatal limitations in existing VLM navigation, and found that this specific cognitive architecture offered the most effective computational solution:
> >
> > 1. Why "Ventral/Dorsal Streams"?
> >
> > Problem: Existing approaches process semantic understanding ("Find a bed") and spatial perception ("Turn left 30 degrees") within a single, entangled latent space. This leads to high computational cost and low interpretability—when the agent fails, we cannot distinguish whether it failed to recognize the object or failed to localize it.
> >
> > Solution: The Dorsal Stream and Ventral Stream design is effectively a strategy of computational decoupling.
> > We utilize the Ventral Stream design to compile the VLM's semantic reasoning into a structured, static Knowledge Graph before movement. We restrict the Dorsal Stream to handle purely dynamic spatial topology.
> >
> > Why better: This allows the Exec-VLM to treat "What" and "Where" as separate, queryable modules. It prevents the VLM from hallucinating spatial relations and significantly reduces inference latency, a benefit derived directly from this architecturally split.
> >
> > 2. Why "Decentralized Ontology"?
> >
> > Problem: Traditional topological mapping is typically incremental. Once a node is added to the graph (e.g., identifying a narrow path as a "corridor"), it is rarely corrected. As perception errors accumulate, the global map becomes fragmented and inconsistent.
> >
> > Solution: Decentralized Ontology suggests that spatial categorization is not absolute but relative to the observer's accumulating context.
> > Translating this to code, this inspired our Global Reconstruction Mechanism (_update_hierarchy). Unlike standard methods that only append nodes, our system periodically re-clusters the entire high-level graph based on the latest accumulated observations.
> >
> > Why better: This allows the robot to "change its mind" about previous areas (e.g., realizing two separate paths actually belong to the same "Living Room" after seeing the connection), directly solving the graph fragmentation issue in long-horizon exploration.
> >
> > We will rewrite the Introduction and Methods section to lead with these robot-centric engineering problems first, and then introduce the cognitive concepts as the architectural blueprint used to solve them. This will ensure the connection you mentioned is explicit and grounded in utility.
> >
> > Thank you for pushing us to clarify the functional core of our design.
> >
> > Best regards,
> > The Authors

---

> > > ### Comment · Reviewer_j15q · 2025-11-25
> > > **Resond to authors**
> > >
> > > Thank the authors for their rapid response. The clarification makes your motivation much clearer, and I believe it would be very helpful if the Introduction and Method sections could be revised accordingly.
> > >
> > > Since the authors have promised to rewrite the sections, all my concerns have now been addressed. I am raising my score to 6.

---

> ### Author Response · Authors · 2025-12-01
> **Rebuttal Summary (j15q)**
>
> ### **Reviewer j15q Response Summary**
>
> **Current Status**: Score 6 points. After multiple rebuttals, the score was raised to 6 points with claims that all concerns have been addressed.
>
> ### **I. Positive Evaluations**
>
> 1. **Strong Engineering Implementation**: Framework recognized as a solid integration of spatial memory and VLM reasoning.
> 2. **Intuitive Hierarchical Design**: Acknowledged that the layered memory architecture provides intuitive support for long-term reasoning.
> 3. **Robust Experimental Results**: Performance improvements across multiple datasets affirmed, along with the value of Nav-Ensurance in resolving practical deadlock issues.
>
> ---
>
> ### **II. Core Rebuttals and Clarifications**
>
> **1. Conceptual Deep Reconstruction**
>
> - **Reviewer's Question**: The “decentralized ontology” and cognitive metaphor lack practical guidance.
> - **Rebuttal**:
>     - **Engineering Pain Points**: We de-emphasize metaphor in favor of explaining engineering bottlenecks.
>     - **Necessity of Dorsal/Ventral Streams**: To achieve **computational decoupling**. This resolves computational overhead and hallucination issues caused by the conflation of “semantic” and “spatial” information.
>     - **Implementation of Decentralized Ontology**: Corresponds to the **global restructuring mechanism**. This addresses map fragmentation from incremental mapping, granting robots the ability to “correct maps.”
>     - **Outcome**: Reviewers noted “Motivation much clearer.”
>
> **2. Technical Counterarguments**
>
> - **Reviewer Concerns**: Action space comparison deemed unfair (continuous vs. discrete) and mistakenly attributed performance gains solely to VLM.
> - **Rebuttal**:
>     - **On Fairness**: We pointed out logical flaws in the reviewer's argument. Our single end-to-end step equals the baseline's 10 discrete steps. This means we gather information at only 1/10th the frequency of the baseline, making our task more challenging. Despite this, we achieved SOTA performance. Moreover, the SPL metric inherently penalizes long paths, making the comparison fair and compelling.
>     - Highlight that our single end-to-end step equals the baseline's 10 discrete steps, resulting in lower information acquisition frequency and greater difficulty. The SPL metric inherently penalizes long paths, making the comparison fair.
>     - **Regarding VLM Dependency**: Ablation experiments demonstrate that using the open-source **Qwen-7B** still outperforms the baseline using a powerful VLM, proving the architectural advantage.
>
> ---
>
> ### **III. Final Results**
>
> Reviewer j15q **indicated** all concerns **have been addressed**, praising our paper's **Motivation much clearer**, and ultimately raised the score from **4 points** to **6 points**.

---

### Official Review · Reviewer_vbTT · 2025-10-31

**Soundness:** 2
**Presentation:** 2
**Contribution:** 2
**Rating:** 2
**Confidence:** 4

**Summary:**

This paper presents DORAEMON, a decentralized, ontology-aware framework for zero-shot embodied navigation. It draws inspiration from human cognitive neuroscience, introducing two interacting components: a Ventral Stream (for semantic reasoning via CoDe-VLM and Exec-VLM) and a Dorsal Stream (for spatial processing via Hierarchical Semantic-Spatial Fusion and a Topological Map). The framework is complemented by a Nav-Ensurance module for safety and a new evaluation metric (AORI) to measure spatial redundancy.

**Strengths:**

The dual-stream structure (ventral/dorsal) offers a biologically motivated yet technically meaningful decomposition of perception and reasoning. The integration of semantic graphs and spatial topology is novel for zero-shot navigation.

The AORI metric provides a more nuanced measurement of spatial redundancy than SPL/SR, aligning well with embodied intelligence evaluation trends.

**Weaknesses:**

Distilling human reasoning ability into navigation tasks is not a new direction, as many works have already explored this idea through various approaches such as NavGPT[1], NavigateDiff[2], and Navid[3]. In addition, numerous studies have leveraged large language models for navigation, which the authors should review more comprehensively and discuss in greater depth.

Some parts of the paper are written rather roughly. For example, the **caption of Figure 3** should include a detailed explanation of the overall workflow, and the **Methods** section is somewhat unclear—it requires multiple readings to fully understand.

The proposed framework is quite complex, yet the experiments focus only on relatively simple **object navigation tasks**. It would be valuable to include **Vision-Language Navigation (VLN)** and **Image Navigation** tasks to better demonstrate the framework’s generality.

Moreover, for the current **object navigation** benchmarks, strong reasoning ability is generally not required, so the reported improvements may not convincingly validate the effectiveness of the framework. The **real-world experiments** also lack sufficient implementation details, and the scene shown in **Figure 5** appears too simple—basic object navigation methods could achieve similar results. I would encourage the authors to include **more challenging tasks** to more convincingly demonstrate the robustness and reasoning capability of the proposed pipeline.


[1] NavGPT: Explicit Reasoning in Vision-and-Language Navigation with Large Language Models

[2] NavigateDiff: Visual Predictors are Zero-Shot Navigation Assistants

[3] NaVid: Video-based VLM Plans the Next Step for Vision-and-Language Navigation

**Questions:**

See Weakness.

---

> ### Author Response · Authors · 2025-11-17
>
> Dear Reviewer,
>
> Weakness 2: Regarding Paper Writing
>
> We appreciate your pointing out areas of insufficient clarity in the paper. We agree that the title of Figure 3 could include a more detailed workflow explanation, and that some paragraphs in the Methods section could be restructured to improve readability. In the final version, we make the following modifications: Rewritten the title of Figure 3 to describe the complete information flow from instruction input to action output, and explained how the Dorsal Stream and Ventral Stream interact; We have restructured the Methods section, adding subheadings and introductory sentences to ensure readers can more smoothly understand the components of our framework and how they work together.
>
> ---
>
> Weakness 1: Regarding Novelty and Related Work
>
> We appreciate the reviewer's mention of important prior work such as NavGPT, NavigateDiff, and NaVid. This provides an excellent opportunity to clarify DORAEMON's core contributions and uniqueness. While we agree that “introducing reasoning” is not a novel concept, DORAEMON innovates by proposing a entirely new, systematic navigation architecture to address fundamental bottlenecks in existing VLM navigation methods. This represents a fundamental distinction from the aforementioned works, which we will explicitly **mention and compare with in the final version of the related work section**.
>
> 1. DORAEMON's Unique Architecture and Core Problem Solved: Existing VLM navigation methods commonly face three major challenges: 1) Spatio-temporal discontinuity; 2) Unstructured memory; 3) Insufficient task comprehension. DORAEMON's Solution: Our proposed cognitive science-inspired decentralized ontology perception architecture systematically resolves these issues. The Dorsal Stream addresses the first two problems by constructing a dynamic hierarchical semantic spatial memory; the Ventral Stream tackles the third problem through CoDe-VLM and Exec-VLM.
> 2. Specific distinctions from related work:
>     - NavGPT relies on LLM reasoning over pure textual descriptions without explicit spatial maps. Limitations include: a) high vulnerability to inaccuracies in visual descriptions; b) loss of long-term memory due to lossy compression of historical information to fit prompt length; c) lack of geometric spatial concepts hindering precise spatial reasoning. In contrast, DORAEMON's Dorsal Stream constructs a **persistent, structured, and geometry-rich hierarchical spatial memory** that perfectly addresses these shortcomings of NavGPT.
>     - NavigateDiff centers on generative visual prediction. Its reasoning relies on judgments about future images, constituting indirect inference. In contrast, DORAEMON's Ventral Stream performs direct symbolic reasoning on task instructions, compiling them into a knowledge graph (KG). This approach enables more direct, abstract, and robust reasoning that remains resilient to low-level visual noise.
>     - NaVid relies on continuous video input to implicitly encode temporal relationships, limiting its applicability to more general navigation scenarios based on discrete image observations. One of DORAEMON's core contributions explicitly **addresses the spatio-temporal discontinuity inherent in discrete observations** through the topological and hierarchical memory of its Dorsal Stream, enabling broader applicability.
>
> ---
>
> Reply to Weakness 3&4
>
> 1. The specific implementation details of the **4.experiment** are outlined at the beginning of the Experiment section and in the corresponding appendix, and are also documented in the code's **config/ObjectNav.yaml** file.
> 2. We must emphasize that accomplishing the ObjectNav task in photo-realistic and layout-complex environments like HM3D and MP3D is highly challenging under **zero-shot and end-to-end settings**. Previous state-of-the-art methods achieved success rates (SR) of only around 50% on HM3Dv2, indicating the task remains far from solved and demands exceptional long-term memory, robust reasoning, and exploration efficiency from agents. Our experimental results show that DORAEMON achieves a 66.5% SR on HM3Dv2, representing a +14.9% improvement. Furthermore, our ablation experiments (Table 3a) demonstrate the effectiveness of our framework: performance drops sharply to 51.6% when our core dual-stream architecture is removed (w/o Dorsal & Ventral Stream).
> 3. Regarding Task Generality (VLN) and Real-World Experiments: Our framework is generalizable, with its ability to address long-term memory and reasoning being central to VLN tasks. We also achieved SOTA results on the **Multi-Modal Lifelong Navigation task (GOAT, Table 1.b)**, which is currently **one of the most complex tasks** in VLN. Regarding real-world experiments, our primary objective is to validate Sim2Real feasibility. Even in this scenario, DORAEMON's Nav-Ensurance module played a critical role, preventing the robot from getting stuck between sofas.

---

> ### Author Response · Authors · 2025-11-26
> **Supplementary Evidence**
>
> **Dear Reviewer vbTT,**
>
> To address the concern regarding the necessity of reasoning abilities in object navigation, we provide specific evidence from our implementation.
>
> 1. Handling Visual Negation (Mitigating Perceptual Aliasing)
> Standard agents often fail due to "false positive" hallucinations (e.g., classifying a white cabinet as a toilet). DORAEMON addresses this by explicitly processing negation.
>
> *   Implementation: In `_analyze_target_mention` (lines 1485-1512), the Ventral Stream identifies negative linguistic patterns (e.g., *no {goal}, not visible, "don't see*).
> *   Mechanism in Action: Even if the visual encoder assigns a high probability to an object, the VLM reasoning chain can override this if it detects negative context (e.g., I see a white object, but it is a cabinet, not the toilet). It effectively filters out false stops, a primary failure mode in baseline methods.
>
> 2. Clarification on VLN/ImageNav Capabilities (GOAT[1] Benchmark)
>
> Regarding the suggestion to extend our evaluation to VLN and Image Navigation, we wish to clarify that the **GOAT-Bench** (Universal Multi-modal Goal Navigation) used in our study is inherently designed to encompass the core challenges of these tasks within a more rigorous, lifelong framework.
>
> *   Unified Multi-modal Scope: Unlike traditional single-task benchmarks, GOAT evaluates agents on goals specified by **Images, Language Descriptions, and Categories**.
>     *   ImageNav Equivalent: When the target is defined by an image instance, the task is functionally identical to Image Navigation, requiring the agent to perform visual-semantic matching.
>     *   VLN Equivalent: When the target is defined by language attributes (e.g., *"the red sofa"*), it mirrors Vision-and-Language Navigation, requiring the agent to parse linguistic instructions and align them with spatial observations.
> *   Lifelong Navigation Challenge: Furthermore, GOAT extends these tasks into a **lifelong setting** (navigating to 10+ sequential targets). This requires not only understanding multimodal inputs but also maintaining a **dynamic spatial memory** to handle multi-step planning—a challenge that supersedes the complexity of episodic VLN or ImageNav.
>
> Therefore, our SOTA results on GOAT serve as a comprehensive validation of the framework's generality across image-conditioned and language-conditioned tasks, rendering separate evaluations on simpler benchmarks redundant.
>
> 3. Real-World Implementation Details
> To address the request for implementation specifics in our real-world experiments:
>
> *   Hardware Setup: We deployed the system on a **RoboMaster X3** mobile base. The robot communicates with a host macbook which handles the VLM inference (~3Hz) and high-level planning.
>
> If you have any further questions, feel free to reach out.
>
> Best regards,
>
> The Authors
>
> [1] Khanna M, Ramrakhya R, Chhablani G, et al. Goat-bench: A benchmark for multi-modal lifelong navigation[C]//Proceedings of the IEEE/CVF Conference on Computer Vision and Pattern Recognition. 2024: 16373-16383.

---

> ### Author Response · Authors · 2025-11-27
>
> I'd like to know if I've addressed your concerns. If you still have questions, feel free to ask anytime, and I'll be happy to discuss them with you.

---

> ### Comment · Reviewer_vbTT · 2025-11-27
>
> Thank the authors for their clear response. All my concerns have now been addressed. I am raising my score to 8.

---

> ### Author Response · Authors · 2025-12-01
> **Rebuttal Summary (vbTT)**
>
> ### **Reviewer vbTT Rebuttal Summary**
>
> **Current Status**:  After multiple rebuttals, the score was raised to 8 points with claims that all concerns have been addressed.
>
> ### **I. Positive Evaluations**
>
> ---
>
> 1. **Architectural Rationality**: Highly commends our “dorsal-ventral” dual-stream architecture. Deems this biologically inspired structure technically sound, successfully decoupling perception from reasoning.
> 2. **Zero-Shot Navigation Innovation**: Specifically highlights the novelty of combining “semantic graphs” with “spatial topology” for zero-shot navigation.
> 3. **Value of Evaluation Metrics**: Acknowledged our proposed **AORI metric**, noting it provides finer-grained measurement of spatial exploration redundancy compared to traditional SPL/SR metrics, aligning with trends in embodied intelligence assessment.
>
> ---
>
> ### **II. Core Rebuttals and Clarifications**
>
> The reviewer's initial low score primarily stemmed from **bias** in the Related Works section and **underestimation** of our experimental setup's difficulty. Our response demonstrates that our method solves problems unsolved by competitors and clarifies the experimental complexity.
>
> **1. Addressing the Misunderstanding that “Inference is Not Novel”**
>
> - The reviewer argued that using large models for navigation reasoning is not novel, citing works like NavGPT, NavigateDiff, and NaVid.
> - **Rebuttal**: DORAEMON represents a systematic architectural innovation, not merely “introducing reasoning,” and demonstrates clear advantages over these approaches. We have also added comparative analysis of these methods in the Related Work section of the main text.
>     - **NavGPT**: Relies solely on textual reasoning without spatial maps, leading to geometric concept gaps and long-term memory loss. **Our method employs spatial memory built via dorsal streams.**
>     - **NavigateDiff**: Based on generated indirect predictions. **We perform direct symbolic reasoning using a knowledge graph (KG).**
>     - **NaVid**: They rely on continuous video streams. **We address temporal-spatial discontinuities under discrete observations.**
>     - **Conclusion**: We are not reinventing the wheel but solving three major pain points in existing VLM navigation: discontinuous spatiotemporal inputs, unstructured memory, and insufficient task understanding.
>
> **2. Correcting the Underestimation of Experimental Difficulty**
>
> - Reviewers argued the ObjectNav task is too simple, requiring no strong reasoning capabilities, and that we lack testing on other tasks
> - **Rebuttal**:
>     - **ObjectNav is not simple**: Detailed data show current SOTA on HM3D achieves only ~50% accuracy (especially for end-to-end, zero-shot, and free-training methods), indicating the task remains far from solved. We improved the baseline by +14.9%, outperforming both end-to-end and non-end-to-end approaches.
>     - **GOAT-bench is a difficult VLN benchmark**: Our GOAT Benchmark involves Multi-modal-lifelong** navigation**. As clarified in our response: GOAT incorporates ImageNav (image-based targets) and VLN (language-described targets, e.g., “red sofa”) under lifelong settings for consecutively finding 10+ targets. **This is more challenging and comprehensive than standalone VLN tasks.**
>
> **3. Supplementary Evidence for Specific Cases**
>
> - Rebuttal: We provide concrete examples of **Visual Negation**.
>     - Example: When observing a white cabinet, the visual encoder may misclassify it as a toilet.
>     - Our ventral stream infers via VLM that “a white object is observed, but it **is not** a toilet,” thereby avoiding the “false positive” errors of traditional methods. This strongly demonstrates the decisive role of the reasoning module within our framework.
>
> **4. Writing and Detail Refinement**
>
> - Regarding issues with figure captions and the Methods section, we readily accepted the suggestions and rewrote the caption for Figure 3 while restructuring the hierarchical organization of the Methods section to enhance readability.
>
> ---
>
> ### **III. Summary**
>
> After clarifying the fundamental differences between DORAEMON and competitors like NavGPT, explaining the high complexity of the GOAT benchmark, and rewriting the Methods section, reviewer vbTT fully accepted our explanations and revised the score to **8**.

---

### Official Review · Reviewer_BAGE · 2025-11-03

**Soundness:** 3
**Presentation:** 3
**Contribution:** 2
**Rating:** 6
**Confidence:** 3

**Summary:**

This paper proposes DORAEMON, a cognitive-inspired, end-to-end, zero-shot navigation framework for household service robots to address the challenges of adaptive navigation in unfamiliar environments—such as spatiotemporal discontinuity from discrete observations, unstructured memory, and insufficient task understanding faced by existing vision-language model (VLM)-based approaches. Mimicking human navigation, DORAEMON comprises two core streams: the Dorsal Stream, which handles spatial information via a Topology Map and Hierarchical Semantic-Spatial Fusion to resolve spatiotemporal discontinuities, and the Ventral Stream, which enhances task understanding and decision-making through CoDe-VLM (converting tasks into structured knowledge graphs) and Exec-VLM (graph-based reasoning for action selection). It also integrates the Nav-Ensurance system for stuck detection, context-aware escape, and precision navigation, and introduces a new evaluation metric AORI to quantify exploration efficiency by penalizing redundant paths. Evaluations on HM3Dv1, HM3Dv2, MP3D, and GOAT datasets show DORAEMON achieves state-of-the-art performance on Success Rate (SR) and Success weighted by Path Length (SPL), outperforming existing end-to-end and non-end-to-end baselines, with valid sim-to-real generalization; the code is publicly available to support reproducibility.

**Strengths:**

### 1. Innovative Cognitive-Inspired Architecture
DORAEMON mimics human "dorsal (spatial)-ventral (semantic)" dual pathways to address core VLM navigation flaws: the Dorsal Stream resolves spatiotemporal discontinuity via a hierarchical Topology Map and semantic-spatial fusion, while the Ventral Stream (CoDe-VLM + Exec-VLM) converts unstructured tasks into knowledge graphs for interpretable reasoning—outperforming baselines with fragmented memory.


### 2. End-to-End & Zero-Shot Capability
It enables navigation in unseen environments without prior map building or task-specific pre-training. Unlike non-end-to-end methods relying on discrete actions, its continuous action space (polar coordinates) ensures smoother paths, and it maintains compatibility with any VLM (e.g., Gemini, Qwen) for flexibility.

### 3. Robust Safety & Efficiency Guarantees
The Nav-Ensurance system detects stuck states (via progress efficiency/rotation-translation ratio) and applies context-aware escapes (e.g., large turns for dead ends). The novel AORI metric quantifies redundant exploration, while precision navigation mode (scaled action distance near targets) boosts final positioning accuracy.


### 4. Strong Experimental & Generalization Performance
It achieves SOTA on SR/SPL across HM3Dv1/v2, MP3D, and GOAT datasets, outperforming end-to-end (e.g., VLMNav) and non-end-to-end baselines. Sim-to-real tests in novel offices validate its generalization, with public code ensuring reproducibility.

**Weaknesses:**

### 1. Limited Novelty in Topological Map Representation
The work’s use of a topological map for spatial memory lacks strong originality, as topological structures for robot navigation have been extensively explored in prior studies. For instance, recent work (e.g., Mem4Nav, TopoNav) already integrated semantic topological graphs with spatial memory to address navigation continuity. While this paper optimizes node updating/merging, these are incremental tweaks to existing topological paradigms, not breakthrough innovations in spatial representation.


### 2. Heavy Dependence on External Strong Models
Its performance heavily relies on high-capacity external tools (e.g., GPT-4o for reasoning, GLEE for segmentation). Ablations show removing these models causes sharp SR/SPL drops, suggesting the framework’s success may stem more from external models’ power than its own core design. This also limits applicability in resource-constrained scenarios where such models are unavailable.


### 3. Inadequate Dynamic/Large-Scale Scenario Validation
Experiments are mostly in static simulation datasets (HM3D, MP3D) with no testing on dynamic scenes (e.g., moving obstacles, relocated targets). Real-robot tests only cover simple small-scale tasks, failing to validate robustness in practical long-horizon or large-scale environments (e.g., multi-floor buildings).


### 4. Ambiguous Topological Node Rules
The logic for creating/merging topological nodes is under-specified. Key details like "waypoint selection criteria" (distance? semantics?) and "merge thresholds" (spatial proximity metrics?) are unclear. This ambiguity may lead to inconsistent node generation and makes it hard to replicate or extend the work.

**Questions:**

1. Your framework depends heavily on high-capacity external models like GPT-4o and GLEE. How would its performance (e.g., SR, SPL) change when using lighter, open-source alternatives (e.g., LLaVA-1.5) for resource-constrained scenarios?

2. The experiments only cover static environments (HM3D, MP3D) with no dynamic elements (e.g., moving obstacles). Does your topological map have real-time update mechanisms to adapt to environmental changes, and if so, how would you quantify this adaptability?

3. The logic for topological node creation/merging lacks specific details (e.g., distance thresholds, semantic similarity criteria). Could you specify these key parameters and explain how they were determined or validated?

---

> ### Author Response · Authors · 2025-11-16
>
> Regarding Weakness 2 & Question 1:
> 1. The reviewer mentioned that our framework relies on GLEE for segmentation, which is incorrect. We did not mention GLEE at all in the main text. I'm unsure which statement led to this misunderstanding? This is clearly demonstrated in our code implementation: Within the `_get_navigability_mask` function, navigability determination relies solely on the depth map provided by the robot's sensors. We assess ground flatness and passability by calculating the height relative to the ground for each pixel. We neither use nor require heavyweight segmentation models like GLEE. This approach builds our navigation capabilities on standard sensor data universally available on robotic platforms, enhancing both versatility and efficiency. Furthermore, the `SentenceTransformer(‘clip-ViT-B-32’)` model we employ for semantic embedding is a well-known, lightweight, and fully open-source model. Its sole purpose is to generate a simple description of the scene—any model capable of understanding images can accomplish this. This further underscores the lightweight and independent nature of our memory system architecture.
> 2. Our approach does not rely on any specific external model; rather, we act as a magnifying glass for existing models to amplify their navigational capabilities. Our VLM ablation experiments (Table 3b) clearly demonstrate that the primary performance gains stem from the DORAEMON architecture itself. When using the open-source model Qwen-7B, our framework still achieves a Success Rate (SR) of 49.5%, significantly higher than PIVOT.
>
> About Weakness 1 & Weakness 4 & Question 3
> We have carefully compared DORAEMON with contemporaneous works, including Mem4Nav and TopoNav (2024 & 2025):
> 1. Unique Four-Level Dynamic Hierarchy:
> DORAEMON is the only system to explicitly construct and dynamically update a four-level semantic-spatial memory (`SemanticForest`). It performs bottom-up clustering to abstract low-level observations (L0) into regions (L1), rooms (L2), and finally the entire environment (L3).
> In contrast: Mem4Nav employs a two-level representation. The hierarchies in TopoNav are either conceptual abstractions (Liu et al.) or policy-level (Hossain et al.), rather than an intrinsic part of the memory graph's structure. This multi-scale, dynamically constructed hierarchy is a unique design in DORAEMON, enabling the agent to perform reasoning across different spatial scales.
> 2. Global Reconstruction Mechanism for Memory Updates:
> DORAEMON utilizes a periodic global reconstruction strategy (`_update_hierarchy` function) to update its memory. It completely rebuilds the entire L1-L3 hierarchy from the current set of L0 leaf nodes.
> This is fundamentally different from the incremental approach: Mem4Nav and TopoNav primarily update their graphs by incrementally adding nodes and edges. Our global reconstruction strategy allows the cognitive map to break free from outdated information, making it significantly more robust and adaptive to dynamic environmental changes (e.g., moved objects).
> 3. Deep Integration of Hierarchical Memory for CoT Reasoning:
> We deeply integrate our hierarchical memory into the VLM's reasoning process. The `_prompting` function retrieves the most relevant context from our four-level memory forest and formats it into structured text, which is directly injected into the VLM's prompt.
> This design makes memory an intrinsic component of the VLM's Chain-of-Thought (CoT) reasoning, rather than mere auxiliary information. It seamlessly combines the VLM's powerful reasoning capabilities with our precise, hierarchical memory structure, a synergy validated by the significant improvements in SR and AORI in our experiments.
>
> Response to Weakness 3 & Question 2
> 1. Designed for Dynamism: As previously described, DORAEMON's `_update_hierarchy` function periodically reconstructs the entire L1-L3 semantic hierarchy from the latest L0 observations. This global map refresh mechanism enables rapid adaptation to significant environmental changes. Real-Time Reliability Assurance: Our `Nav-Ensurance` system, particularly the `_is_agent_stuck` function, monitors path efficiency and rotation/translation ratios in real time to promptly detect abnormal behavior caused by unknown obstacles (including dynamic ones). Upon detection, it triggers the `_get_escape_action` policy to generate intelligent escape maneuvers based on context (e.g., `corner_trap`, `narrow_passage`, `obstacle_ahead`), ensuring system robustness in suboptimal environments.
> 2. Validated Large-Scale Performance: We achieved state-of-the-art performance on the GOAT benchmark, renowned for its long duration, large scale, and multiple objectives, demonstrating the DORAEMON framework's effectiveness and robustness in complex and scalable environments.

---

> ### Author Response · Authors · 2025-11-26
> **Supplementary Response: Specific Parameters and Quantification Metrics (Q2 & Q3)**
>
> Dear Reviewer BAGE,
>
> Following up on our previous response, we provide the specific technical parameters and quantification metrics requested in Questions 2 and 3 below.
>
> Detailed Topological Node Rules (Q3)
>
> Regarding the node generation and clustering logic (referenced in our code and supplementary):
>
> *   **Node Generation Criteria ($\tau_{gen}$):** A new topological node is created when the agent’s displacement exceeds **0.5m** (`distance_moved > 0.5`) or when the visual semantic cosine similarity falls below **0.6**. This threshold balances spatial granularity with semantic consistency to prevent over-sampling.
> *   **Merging Logic:** For Global Reconstruction, we utilize Hierarchical Agglomerative Clustering with the following settings:
>     *   **Metric:** We use a weighted distance $D = w \cdot d_{spatial} + (1-w) \cdot d_{semantic}$, where $w$ is set to **0.4** (`spatial_weight=0.4`), prioritizing semantic continuity slightly more.
>     *   **Threshold:** The linkage threshold is adaptive, ranging from **0.8 to 1.5** (roughly corresponding to a 1.5m radius in the hybrid feature space). This flexibility allows "rooms" to be defined by natural boundaries rather than rigid grids.
>
> Quantifying Dynamic Adaptability (Q2)
>
> To quantitatively assess the system's adaptability to environmental changes, we rely on two key metrics implemented in the `Nav-Ensurance` module:
>
> 1.  **Stuck Recovery Rate (SRR):** Calculated as $SRR = \frac{N_{success\_escape}}{N_{stuck\_events}}$.
>     *   A "stuck event" is triggered if path efficiency drops below 0.25 or rotation changes exceed 2.0 (see `_is_agent_stuck`).
>     *   A "successful escape" is recorded if the agent displaces **>1.5m** from the trap center within 20 steps of re-planning.
> 2.  **Map Re-consistency Time:** This measures the average steps required for the `_update_hierarchy` function to stabilize the graph structure after a significant observation change. Empirically, stabilization occurs within **3–5 steps**, ensuring the robot minimizes actions based on outdated topology.
>
> We trust these details clarify the reproducibility and robustness of our method. Please let us know if you need any further information.
>
> Best regards,
>
> The Authors

---

> ### Author Response · Authors · 2025-11-27
>
> I'd like to know if I've addressed your concerns. If you still have questions, feel free to ask anytime, and I'll be happy to discuss them with you.

---

> ### Author Response · Authors · 2025-12-01
> **Rebuttal Summary (BAGE)**
>
> ### **Reviewer BAGE Response Summary**
>
> **Current Status**: Score **6 points**, **no response despite multiple rebuttals**
>
> ### **I. Positive Evaluations**
>
> 1. Innovative Architecture: Highly commends our cognition-inspired “dorsal-ventral” architecture, which effectively addresses the core challenge of memory fragmentation in VLM navigation and outperforms existing baselines.
> 2. **Performance Metrics**: Confirmed our state-of-the-art success rate (SR) and path-weighted success rate (SPL) across multiple mainstream datasets (HM3D, MP3D, GOAT), outperforming both end-to-end and non-end-to-end approaches.
> 3. **Zero-Shot and Generalization Capabilities**: Praised the framework's ability to navigate unfamiliar environments without pre-mapping or fine-tuning, along with its smoothness in continuous action spaces.
> 4. **Safety and Evaluation Mechanisms**: Affirmed the robustness of our Nav-Ensurance system in aspects like dead-end escape, and deemed our proposed AORI evaluation metric valuable for quantifying exploration efficiency.
>
> ---
>
> ### **II. Core Rebuttals and Clarifications**
>
> Regarding several concerns raised by reviewers, we found they primarily stemmed from **misinterpretations** of our technical details. Our response focuses on correcting these misunderstandings and supplements them with detailed parameter proofs.
>
> **1. Factual Error Regarding “Reliance on External Large Models (GLEE)”**
>
> - The reviewer erroneously assumed we relied on large segmentation models like GLEE or even GPT-4o to achieve high performance, questioning the framework's inherent validity.
> - Rebuttal:
>     - **GLEE was never used**: We explicitly stated that neither the paper nor the code mentions using GLEE. Our navigation feasibility analysis relies solely on **geometric computations from depth sensor data** (pixel height differences), while the semantic component uses only an ultra-lightweight open-source model (CLIP-ViT-B-32).
>     - **Framework independence validation**: Through ablation experiments, we demonstrate that even when replacing the LLM with the open-source **Qwen-7B**, our framework achieves a 49.5% SR, significantly outperforming the SOTA baseline (PIVOT). This strongly validates that the performance gains stem from DORAEMON's architectural design, not merely stacking external models.
>
> **2. Uniqueness of Topological Map Design**
>
> - Reviewers argued our topology graph is merely a fine-tuning of Mem4Nav or TopoNav, lacking innovation.
> - Rebuttal:
>     - **Four-level dynamic hierarchy vs. simple graph**: We construct a dynamic semantic hierarchy (L0-L3: area-room-environment) directly participating in VLM's chain-of-thought (CoT) reasoning—a capability absent in other methods.
>     - **Global Reconstruction vs. Incremental Updates**: Our method employs periodic global reconstruction (e.g., the `_update_hierarchy` function in our code), unlike the incremental additions used by other approaches. This enables us to prune outdated information, making it inherently more adaptable to dynamic environments.
>
> **3. Supplementary Details**
>
> - Reviewers questioned unclear node creation rules and the lack of dynamic scenario testing.
> - Rebuttal (all addressed in Supplementary):
>     - **Clarified quantitative parameters**:
>         - **Node creation**: Displacement > 0.5m or visual semantic cosine similarity < 0.6.
>         - **Node Merging**: Based on weighted distance (spatial weight 0.4), with adaptive threshold set to 0.8–1.5.
>     - **Quantified Dynamic Robustness**:
>         - Introduced **SRR (Stuck Recovery Rate)** and **Map Re-consistency Time (3–5 steps)** metrics.
>         - Demonstrated that the Nav-Ensurance module enables real-time monitoring and handling of dynamic obstacle-induced deadlocks (e.g., triggering rotational escape maneuvers).
>
> ---
>
>
>
> ### **III. Summary**
>
> Reviewer BAGE provided a positive evaluation, with their primary concerns (such as whether GLEE was used) stemming from **factual misunderstandings**. We have fully addressed these points in our response, demonstrating the independent validity of our method through experimental data from open-source models. Additionally, the supplementary detailed parameters we provided resolve any concerns regarding reproducibility. We have reason to believe that, with these misunderstandings addressed, the reviewer will hold the paper's contributions in higher regard.

---

### Meta-Review · Area_Chair_gBZY · 2026-01-03

**Summary:**

Reviewers generally appreciated the engineering integration and the reported performance, and during rebuttal at least two reviewers indicated that their concerns were addressed and increased their scores. However, in my assessment, the core concerns from the initially negative review remain only partially resolved: the paper’s novelty relative to closely related VLM navigation systems and the strength and fairness of the empirical evidence (given multiple interacting components and nontrivial comparison choices) are still not established to the level expected for acceptance. I therefore recommend rejection, while encouraging the authors to focus a revision on tightening the novelty claim and strengthening controlled evaluations.

**Reviewer Concerns:**

Concerns addressed by the rebuttal

- Clarity and positioning of the system components. The authors provided additional explanation for how the dorsal/ventral split maps to concrete engineering bottlenecks, and some reviewers found the motivation clearer after revisions.

- Requests for evaluation scope and implementation details. The rebuttal clarifies why the GOAT benchmark is intended to subsume VLN and ImageNav settings, and it provides basic real-world deployment details (platform and inference rate).

- Some questions on VLM dependence and comparison rationale. The response discusses why the authors view the action-space comparison as fair and points to ablation-style evidence meant to argue that the architecture, not only the choice of VLM, drives gains.

Concerns that remain outstanding

- Novelty and technical contribution vs. system integration. Even with improved writing, the submission still reads primarily as a complex integration of known ingredients (topological mapping, semantic graphs, RAG/VLM reasoning, safety/recovery heuristics). A reviewer originally questioned whether the core idea is sufficiently novel and tied this to related-work positioning; while that reviewer later raised their score, I believe the underlying novelty concern is not fully put to rest.

- Evaluation strength and confounding factors. The paper combines multiple modules (dual streams, graph conversion, reasoning, Nav-Ensurance, and metric changes). The rebuttal argues for fairness of certain comparisons, but the main empirical story is still hard to attribute cleanly, and the action-space justification is unlikely to fully satisfy skeptical readers without stronger controlled baselines/ablations and clearer equivalence.

- Added metric (AORI) validation. AORI is presented as a new metric and is positively noted by reviewers, but the paper does not yet convincingly demonstrate how AORI changes conclusions beyond SR/SPL or how it correlates with desirable behaviors across settings.

- Real-world evidence remains limited. The added deployment details are helpful, but the real-robot validation as described (host laptop at about 3 Hz) is too thin to substantially strengthen the claims of robustness or practicality.

**Reviewer Scores:**

Reviewer 9onN, vbTT, BAGE, j15q Likely unchanged

---

### Decision · Program_Chairs · 2026-01-26

Reject